# Discrete Spatial Diffusion: Intensity-Preserving Diffusion Modeling

**Javier E. Santos**\*  **Agnese Marcato**\*  **Roman Colman**\*  **Nicholas Lubbers**\*

**Yen Ting Lin**\*

## Abstract

Generative diffusion models have achieved remarkable success in producing high-quality images. However, these models typically operate in continuous intensity spaces, diffusing independently across pixels and color channels. As a result, they are fundamentally ill-suited for applications involving inherently discrete quantities such as particle counts or material units, that are constrained by strict conservation laws like mass conservation, limiting their applicability in scientific workflows. To address this limitation, we propose Discrete Spatial Diffusion (DSD), a framework based on a continuous-time, discrete-state jump stochastic process that operates directly in discrete spatial domains while strictly preserving particle counts in both forward and reverse diffusion processes. By using spatial diffusion to achieve particle conservation, we introduce stochasticity naturally through a discrete formulation. We demonstrate the expressive flexibility of DSD by performing image synthesis, class conditioning, and image inpainting across standard image benchmarks, while exactly conditioning total image intensity. We validate DSD on two challenging scientific applications: porous rock microstructures and lithium-ion battery electrodes, demonstrating its ability to generate structurally realistic samples under strict mass conservation constraints, with quantitative evaluation using state-of-the-art metrics for transport and electrochemical performance.

## 1 Introduction

Diffusion-based generative models have emerged as powerful tools for high-quality image generation [36, 66, 69]. Typically, these models inject noise into the images, then learn to reverse this noise-adding process to recover meaningful structure. In most frameworks, this is based on an Itô Stochastic Differential Equation (SDE) with Gaussian noise. While effective for many vision tasks, these approaches inherently assume continuous pixel intensities, which can cause difficulty when dealing with the discrete nature of many datasets. Tasks beyond vision, such as those in the physical sciences have many applications which require discrete physical quantities, such as particle counts in a simulation, or phases in materials microstructure. Conservation of total quantities is critical for scientific applications, and so generative modeling which can operate under constrained, discrete pixel intensities would enable scientifically grounded and physically consistent synthesis. Such a capability might also prove useful within vision tasks, such as inpainting and super-resolution.

Scientific and engineering studies of the natural world using computational techniques often involve discrete variables in space and/or time. On microscopic scales, everyday materials exhibit extremely complex structural patterns which encode the history of their formation, and play a large role in how the material functions on a macroscopic level. An important and wide-reaching field of study is materials microstructure, which is used in materials design [35], forensic analysis, hydrology [7],

---

\*Los Alamos National Laboratory

39th Conference on Neural Information Processing Systems (NeurIPS 2025).

energy storage [65], and even medicine, such as in studies of bone structure [53]. For example, crystal grain shapes can give rise to complex stress patterns which affect the yield strength of a metal [10]. Materials microstructures are frequently represented by a small number of discrete phases that describe their underlying chemical structures. In sandstone, for example, the overall arrangement of nanocrystals is highly disordered, and gives rise to complex pore structures, through which subsurface water flows. This microstructure has an enormous influence on the rate of transport of fluids and contaminants [6]. Microstructure of electrodes is also known to have an immense impact on the characteristics of electrochemical devices [58]. Small changes in thermodynamic properties can cause drastic changes in microstructure, such as in stainless steels [79], requiring the study of microstructure as a function of phase contents. Furthermore, gathering real-world data is often complex and expensive; decades of work have been applied to computational modeling of the generation and consequences of microstructure prior to the widespread popularization of machine learning [72]. While diffusion models have been constructed for microstructure [4, 25, 37, 46, 47, 52, 80], they have not handled exact porosity (intensity) constraints.

In this work, we introduce Discrete Spatial Diffusion (DSD), a discrete-state Markov chain-based diffusion framework in which the forward process redistributes discrete units of intensity in space (Fig. 1c). Unlike previous diffusion models, DSD *exactly* preserves total intensity throughout both the forward and reverse phases, ensuring that global properties—such as mass fractions—are exactly conserved by representing them in terms of conserved particles. We demonstrate that DSD enables scientific applications and can extend the capabilities of conventional image processing tasks, like image generation and inpainting in discrete domains. By directly modeling discrete transitions, DSD enables generative modeling under conservation laws, allowing models that specialize for constrained conditions in scientific applications and beyond.

## 2 Background

Among the body of literature on generative diffusion models, originating from the pioneering work of Sohl-Dickstein et al. [66], the most relevant to our work fall into two broad categories: (1) those employing discrete-state Markov chains to introduce noise in the forward process [3, 11, 39, 51, 63, 64, 70], and (2) those incorporating spatial dynamics into the forward diffusion process [5, 39, 61].

Generative diffusion modeling based on discrete-state Markov chains has become an active area of research in recent years. Early work, such as Austin et al. [3], Hoogeboom et al. [39], introduced discrete-state and discrete-time Markov chains as an alternative to the Gaussian noise used in conventional diffusion models [36, 66, 69]. Campbell et al. [11] generalized these formulations to a continuous-time framework, providing a more rigorous theoretical foundation for discrete-state generative diffusion modeling. Santos et al. [63] employed operator algebraic analysis to formally establish the existence of the reverse-time dynamics and derived the stochastic generator for arbitrary discrete-state Markov processes. Similar formulations were independently developed by Sun et al. [70] and Lou et al. [51], with an emphasis on defining and estimating score functions for discrete-state systems. The Markov process operates in intensity space in all the aforementioned diffusion models, treating each pixel as an independent stochastic process (Fig. 1(a): Gaussian; Fig. 1(b): Discrete).

This study focuses on a spatially correlated process for generative modeling for two reasons: (1) for structured images, it is more natural to incorporate spatial correlations into the generative process, and (2) spatially decorrelated noise makes it difficult to preserve total intensity. A spatially correlated approach has been explored for continuous systems. Cold Diffusion [5] introduced a deterministic blurring transformation, where image degradation follows a predefined forward process, and reconstruction is learned as an inverse mapping. However, lacking a probabilistic latent distribution (as in VAEs [42]), Cold Diffusion is not a true generative model. Inverse Heat Dissipation Model (IHDM, Rissanen et al. [61]) uses the heat equation as a corruption model. Since the heat equation is deterministic and reversible (except that the homogeneous solution at $t \to \infty$ is singular), a naïve inversion would again result in deterministic reconstructions. Uncorrelated Gaussian noise was added to the heat equation to overcome this limitation, relaxing the deterministic process into a probabilistic Itô diffusion. Later, Blurring Diffusion Model (BDM, Hoogeboom and Salimans [38]) recognized that IHDM could be recast as a Gaussian diffusion model in the spectral domain. BDA extended IHDM and achieved state-of-the-art generative performance, validating the hypothesis that spatially structured diffusion processes can enhance image generation. Nevertheless, the probabilistic

formulation of IHDM and BDM only preserves intensity on average, not exactly per-sample, and their continuous-state nature makes it difficult to apply to discrete datasets.

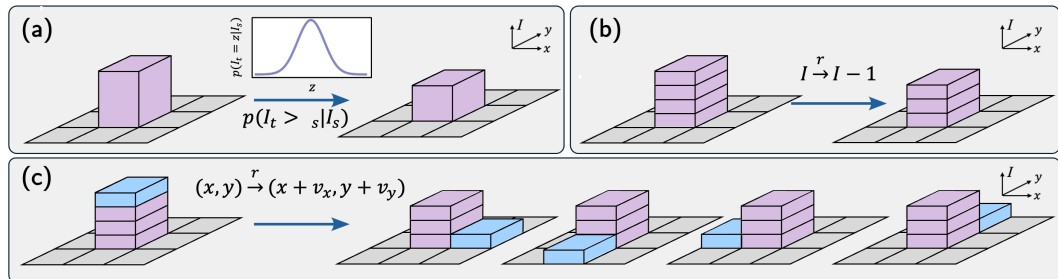

Figure 1: Schematic diagrams illustrating how intensity is modeled in different diffusion frameworks. **(a)** Gaussian Diffusion relies on the Ornstein–Uhlenbeck process in continuous intensity space. **(b)** Prior discrete-state diffusion models apply a discrete-state Markov process to independent pixel intensities. **(c)** Discrete Spatial Diffusion (this work) relies on a Markov jump process of the intensity units over a discrete spatial lattice, redistributing particles while exactly conserving total intensity per color channel.

Our goal of generating samples with exactly conditioned total intensity aligns with conditional diffusion modeling. However, existing approaches all rely on some degree of approximation. Song and Ermon [68] proposed a simple conditional sampling method by passing class labels into the neural network during training, but this does not guarantee exact enforcement of the condition in generated samples. A more structured approach was introduced by Chung and Ye [15], Chung et al. [17, 18], which interleaved projection steps with diffusion sampling to enforce linear constraints in image generation. However, these projections disrupt the exactness of the forward corruption and reverse inference dynamics [2, 11, 63], leading to a mismatch between the projected and true data manifolds. To address this, Chung et al. [16] eliminated projection steps but instead relaxed deterministic constraints into a probabilistic formulation via a noisy measurement model. However, this method does not apply to deterministic constraints, as it becomes singular in the limit of zero measurement noise. An alternative approach leverages Bayes' theorem for a posteriori conditional sampling, that is, $p(S|C) \propto p(S)p(C|S)$, where "S" stands for samples and "C" for condition(s). Because $p(S)$ is given by a trained unconditional diffusion model, conditioning can be performed if one has $p(C|S)$, which is however intractable for arbitrary data distributions[2]. Existing methods approximate this term crudely or by training a separate classifier as in Song et al. [69], or by a Gaussian approximation with moment-matching as in [23, 27]. None of these methods guarantees that the generated samples are exactly conditioned.

The principal contribution of our work is that it provides a new capability for diffusion models to preserve intensity exactly in a fully discrete-state context. The approach is based entirely on how the diffusion process is built, and how the model is trained; it is readily usable with existing diffusion model neural network (NN) architectures. To our knowledge, this is the first diffusion model to incorporate spatially correlated noise, achieved through a stochastic jump process that allows units of intensity to perform a random walk. This fact also demonstrates that more complex noise processes can themselves be tractable. We furthermore demonstrate that such a model is effective for conventional image synthesis tasks. The relevance and power of the approach is then demonstrated through application to scientific data in the field of materials microstructure, where the ability to generate complex data-driven images with constrained total intensity is key to physically meaningful generative modeling and scientific reliability. To clarify DSD's advantage in scientific settings, we explicitly compare to traditional frameworks in Appendix E.

---

[2]It is challenging because the constraint is imposed on the *final samples at the end of the inference*, but the conditioning "S" are *samples generated during the inference*.

## 3 Methods

### 3.1 Corruption Process

In this manuscript, we adopt the language of image processing and consider 2-dimensional images, although the application context and spatial dimensionality of the data are not constrained by the mathematical framework provided here. We treat a digital image with discretized intensity values $I_{x,y,c} \in \mathbb{Z}_{\geq 0}$ at pixel $(x,y) \in \{1,\dots,W\} \times \{1,\dots,H\}$ in color channel $c \in \{1,\dots,C\}$. Within the DSD framework, the intensity values in the image are treated as a spatially organized collection of particles, with one particle for each intensity unit. Below, we will interchangeably use "particles" and "intensity units" to denote these fundamentally discrete units. Specifically, $I_{x,y,c} = n$ implies $n$ particles of type $c$ at location (x,y), and the total number of particles of the system is $\sum_{x=1}^{H} \sum_{y=1}^{W} \sum_{c=1}^{C} I_{x,y,c}$. In the forward stochastic process with the time parameter $t$, each of the particles in the system *independently* performs a continuous-time and discrete-state random walk:

$$(x,y,c) \xrightarrow{r} (x + \nu_x, y + \nu_y, c), \tag{1a}$$
$$\nu := (\nu_x, \nu_y) \in \{(1,0), (-1,0), (0,1), (0,-1)\} \tag{1b}$$

where $r$ is the transition rate of the particle jumping to one of their nearest neighbors, and $\nu$ is a set of four directions the particles can hop to their nearest neighbors. A schematic diagram is shown in Fig. 1c. Note that the particles perform jumps in the $(x,y)$ space at random times, but do not change their color coordinate $c$. Because of this, the forward process conserves the total number of particles $\sum_{x=1}^{H} \sum_{y=1}^{W} I_{x,y,z}$ in each color channel independently. We impose either no-flux boundaries, such that the transition rate for jumps out of the image domain are zero, or periodic boundaries, so that a jump to $x = W + 1$ becomes a jump to $x = 0$, vice-versa, and analogously for $y$.

We refer to the spatial hopping process (Eq. (1)) as the *Discrete Spatial Diffusion* (DSD), noting the "discreteness" refers to both the discretized intensity units and the discreteness of the spatial lattice $\{1,\dots,W\} \times \{1,\dots,H\}$ where the particles are allowed to reside. DSD, as well as similar discrete-state random walks, have been extensively studied in non-equilibrium statistical physics and stochastic processes (Van Kampen [75], Gardiner [28], Giuggioli [33] and references therein). The evolution of the probability distribution of the single random walk in the continuum space limit, under the appropriate scaling of the transition rate [26], converges to the Fokker–Planck Equation (FPE, Risken [60], Van Kampen [75]), which is mathematically identical to the heat equation. Because of the duality between the probabilistic FPE and the deterministic heat equation [44], DSD can be considered as a microscopic description of the macroscopic heat dissipation that inspires IHDM and BDM. Notably, the correlated noise is built in DSD, in contrast to the heuristic addition of uncorrelated Gaussian noise in IHDM and BDM. Fig. 2 illustrates the application of DSD to a sample image. Due to the stochasticity of the random jumps, the limiting behavior ($t \to \infty$) of this process is a random configuration with no discernible structure or similarity to the original spatial organization aside from the conserved global particle counts in each color channel.

We use $(X_t, Y_t, C_t)$ to denote the random process in $(x,y,c)$ space, and $(x_0, y_0, c_0)$ are the initial condition of a specific particle. We use $I_t$ to denote the randomly corrupted image at the time $t$, where $[I_t]_{x,y,c}$ is the total number of particles at $(x,y)$ in color channel $c$. The process can be represented in these two dual representations: with $(X_t, Y_t, C_t)$ the process is formulated in the frame of a moving particle (the Lagrangian frame), and with $I_t$ the process is formulated as a histogram in space-time (the Eulerian frame). Below, we will use these two representations interchangeably.

The forward solution and the transition probabilities $p_t(x,y,c|x_0,y_0,c_0) := \mathbb{P}\{X_t = x, Y_t = y, C_t = z | X_0 = x_0, Y_0 = c_0, C_0 = c_0\}$, can be computed by integrating the Master Equation [28, 75, 77]. This corresponds to exponentiating the Markov transition matrix of the process defined in Eq. (1). While the matrix exponential required numerically for no-flux boundaries is expensive, the solution can be stored and reused to corrupt images and to compute the reverse-transition rates (see Sec. 3.3) for learning. When periodic boundary conditions are imposed, the transition matrix is diagonal in the discrete Fourier space, facilitating the efficient computation of $p_t(\cdot|\cdot)$ (see Sec. C).

### 3.2 Designing Noise Schedules by Structural Similarity Index Metric (SSIM)

Since the corruption process is time-homogeneous (1), the noise applied to each particle remains constant over time. However, it has been shown that inhomogeneous noise schedules can facilitate

learning [55]. We use the formulation of a recent study [62] identified the unique correspondence between non-uniform observation times in a homogeneous Ornstein–Uhlenbeck process [73] and noise schedule in conventional diffusion models [36, 67]. We follow the same philosophy as Santos and Lin [62] to construct a sequence of observation times $t_0 = 0 < t_1 < t_2 < \ldots < t_T = 1$, at which we will generate random samples for learning. Here, $T$ denotes the total number of discrete time steps used to generate corrupted sample images during training.

We adopt a heuristic approach to construct the discrete times. The idea is to use a metric to quantify how much the "quality" of the images has been degraded up to time $t$, and we aim to design $t_k$'s such that the metric degrades from $k = 0$ to $k = T$ as evenly as possible. We chose the Structural Similarity Index Metric (SSIM, Wang et al. [76]) between the corrupted image and the original image. We generalize a generic monotonic relation between $k$ to $t_k$ proposed by Santos et al. [63]:

$$\Phi\left(e^{-\tau_2 t_k}\right) \triangleq \frac{(k-1)\,\Phi\left(e^{-\tau_2}\right) - (T-k)\,\Phi\left(e^{-\tau_1}\right)}{T-1}, \tag{2}$$

where $\Phi(p) := \log p/(1-p)$ is the logit function, $\tau_1$ and $\tau_2$ are parameters used to construct the observation times. Note that $t_T = 1$ in the above parametrization. Specifically, we tune $\tau_1$, $\tau_2$ and the unit transition rate $r$ in process (1), using a subset of training samples, aiming to cover an even degradation of the SSIM throughout observation times. We use $\tau_1 = 7.5$ and $\tau_2 = 2.5$, and $r = 120\text{-}160$, chosen to equalize SSIM degradation over timesteps. We remark that the choice of the functional form in Eq. (2) is picked empirically; we treat Eq. (2) as a versatile monotonic fitting function, whose corresponding SSIM degradation is more symmetric than polynomial and cosine schedules [55] for the DSD process (see Appendix Fig. 6).

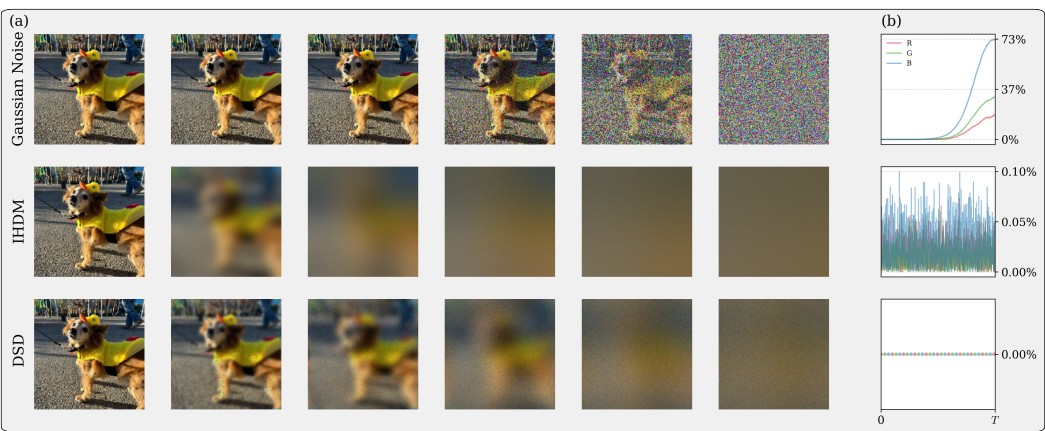

Figure 2: **(a)** The forward processes for Gaussian Diffusion [36], Inverse Heat Dissipation Model [61], and Discrete Spatial Diffusion (ours) applied on an image, sampled at discrete times. **(b)** Percentage change in intensity relative to the original image under the forward process.

### 3.3   Reverse-time process

Following the process formalism from [11, 63], there exists a reverse-time process that evolves in opposite time and whose joint probability distribution is identical to that of the forward process (1). Specifically, the reverse-time process corresponding to process (1) is:

$$(x, y, c) \xrightarrow{r\frac{p_t(x+\bar\nu_x, y+\bar\nu_y, c|x_0, y_0, c_0)}{p_t(x, y, c|x_0, y_0, c_0)}} (x + \bar\nu_x, y + \bar\nu_y, c), \tag{3}$$

where the admissible reverse-time transitions $\bar\nu = (\bar\nu_x, \bar\nu_y) :=\in \{(-1, 0), (1, 0), (0, -1), (0, 1)\}$ are the reversed direction of the forward jumps ($\bar\nu_x = -\nu_x$, $\bar\nu_y = -\nu_y$). The framework ensures the same boundary condition to be imposed (no-flux or periodic, according to the forward process). We note that the reverse-time process, and therefore the generated images, also conserve the total particle number per color channel.

We note that the reverse transition rate depends on both the initial condition $(x_0, y_0, c_0)$ of a particle and the forward solution $p_t(x, y, c|x_0, y_0, c_0)$, $\forall (x, y, c)$. This is analogous to conventional diffusion

models, where either the reverse-time drift [36, 66] or the score function [69] formally depend on the initial sample and the solution of the forward process. However, during the inference, the initial particle configuration is not known, and as such, we train an NN to learn the reverse transition rates using samples $I_t$ generated from the forward process (1) at $t > 0$. Additionally, the particles are indistinguishable, but the rate prescribed in Eq. 3 is *per-particle*, raising the question: what is the appropriate *per-pixel* reverse transition rate that the NN ought to model? This question can be answered by performing the survival analysis of the many-particle system in light of the independence of particle motion; see Appendix A for a derivation. The analysis shows that the reverse transition rate of the first jump of $n = [I_t]_{x,y,c}$ particles is simply the sum of the instantaneous transition rates:

$$\bar{r}_{\bar{\nu},x,y,c} = r \sum_{i=1}^{n} \frac{p_t(x + \bar{\nu}_x, y + \bar{\nu}_y, c | x_0^{[i]}, y_0^{[i]}, c_0^{[i]})}{p_t(x, y, c | x_0^{[i]}, y_0^{[i]}, c_0^{[i]})}, \tag{4}$$

Eq. 4 prescribes the rate for *the first* of all the particles (which is $[I_t]_{x,y,c}$) to jump to one of its neighboring pixels. Intuitively, this can also be derived by combining the first-reaction method [30] and inhomogeneous Poisson process (e.g., see Corbella et al. [20]). Eq. 4 also prescribes the rate that the NN will model. This rate is implicitly time-dependent through the dependence on the forward solution $p_t$, similar to standard continuous-time diffusion models.

## 3.4 Loss functions

Our goal is to provide the corrupted images $I_t$ at a sampled time $t > 0$ to a neural network (NN) and to train it to predict the reverse-time transition rates $\bar{r} \in \mathbb{R}_+^{4 \times H \times W \times C}$ of Eq. 4. The first dimension (of size four) represents rates for the four nearest-neighbor transitions. We denote the NN modeled rates as $\bar{r}^{NN}(I_t, t)$ with noisified image $I_t$ at time $t$.

We use two approaches to formulate the loss function. The first and more commonly used approach adopts a metric to heuristically match the NN prediction and the ground truth, as in DDPM [36], score-matching [69], and flow-matching [49]. We extend these schemes to *rate-matching*, where we minimize a chosen norm of the difference between the predicted and true rates $\bar{r}^{NN}$ and $\bar{r}$. For example, for using L1, a loss $\mathcal{L}$:

$$\mathcal{L}_{\text{rate-matching}} = \mathbb{E}_{I_{t_k},k} \left[ \text{mean}(|\bar{r}^{NN} - \bar{r}|) \right]. \tag{5}$$

Here, $k \in \{1, \dots T\}$ is uniformly sampled, $I_{t_k}$ is drawn from the random process (1) at the sampled times, $\bar{r} = \bar{r}_{\bar{\nu},x,y,c}(I_t, t | I_0)$ is the true reverse-time transition rate (4), $\bar{r}^{NN} = \bar{r}^{NN}_{\bar{\nu},x,y,c}(I_t, t)$ is the NN-predicted reverse-time transition rate, and the mean is over all the indices $(\bar{\nu}, x, y, c)$. The second and more principled approach is through minimization of the negative log-likelihood $L$ of the NN-induced process to predict the analytical reverse-time process [11, 63, 66]:

$$\mathcal{L}_{\text{likelihood}} = -\log L = -\mathbb{E}_{I_t} \left[ \int_0^\infty \sum \left( \bar{r}^{NN} - \bar{r} \log \bar{r}^{NN} \right) dt \right]. \tag{6}$$

Because we only observe the process at discrete times prescribed in Eq. (2), we approximate the continuous-time integration above by

$$\log L = \mathbb{E}_{I_{t_k},k} \left[ (t_k - t_{k-1}) \sum \left( \bar{r}^{NN} - \bar{r} \log \bar{r}^{NN} \right) \right], \tag{7}$$

where we again take expectation over randomly sampled $t_k$ and $I_k$. In this study, we experimented with both loss functions and observed no noticeable difference, giving evidence that the DSD forward process (1) is not sensitive to the choice of the loss function. This stands in contrast to Gaussian diffusion models, where training loss choice significantly impacts performance Ho et al. [36], which used the heuristic approach to improve over Sohl-Dickstein et al. [66], which adopted the second approach. We focus on learning the transition rates of the reverse-time dynamics, which is distinct from the ratio-matching approach [51, 70] which focuses on learning the probability distribution $p_t(\cdot)$, although a similar formulation ("implicit score entropy") proposed by [51] can be regarded as the process likelihood (7) first proposed in Santos et al. [63]. Algorithm 1 describes the DSD training pseudocode.

### 3.5 Sampling with an Adaptive Time Stepping using the Courant–Friedrichs–Lewy condition

Once trained, the neural network will predict reverse-time rates (4) from the configuration of system $I_t$, at time $t \geq 0$. Because the reverse rates are time-dependent, one could generate the exact sample paths of the inhomogeneous Poisson process by integrating the survival function of the first reaction on each pixel in each color channel (see e.g., algorithms reported in Corbella et al. [20]). However, this approach is not computationally efficient, so we resort to $\tau$-leaping [31], an integrator that has been adopted by essentially all continuous-time and discrete-state diffusion models [11, 59, 63, 78], analogous to the Euler's method for ordinary or partial differential equations and Euler–Maruyama for Itô SDEs. The central idea of $\tau$-leaping is to approximate the reverse-time transition rates $\bar{r}$ as a fixed constant in a small enough window $(t - \tau, t)$, assuming the time-dependent rates change slowly in the period, a condition often termed as the "leap condition" [12, 31]. With this assumption, the original $\tau$-leaping algorithm by Gillespie [31] generates Poisson random numbers to update the system's discrete states. However, this approach could sometimes lead to a negative population of particles, which cannot happen in the process, due to violations of the leap condition. Mitigation strategies exist [12, 13, 32], however, some of them are limited to small reaction networks and not suitable for the DSD sampling task, which involves a very large number $(4 \times H \times W \times C)$ of transition rates to estimate.

As such, we propose a more efficient (but arguably cruder) approach to select the step size $\tau$ adaptively. Our idea is to combine the binomial $\tau$-leaping [14, 71] and the Courant–Friedrichs–Lewy (CFL) condition [21] to conservatively determine the adaptive step size $\tau$. Specifically, since the jump scale is fixed at the pixel length scale, the timescale $\tau$ fully determines the CFL condition. The idea is to choose a $\tau$ such that the CFL number is fixed throughout the inference[3]. To achieve this, we compute the reverse-time transition rates $\bar{r}_{\bar{\nu}}$ for each pixel in each channel, noting that the probability of a particle in that channel will jump to one of its neighboring locations is $\bar{r}_{\bar{\nu}}\tau$. Then, we determine $\tau$ by fixing the largest probability across all the pixels and color channels at a constant. Algorithm 2 describes the DSD inference pseudocode.

## 4 Computational Experiments

We employ the Noise Conditional Score Network (NCSN++) [1, 69] with two modifications: the final convolutional layer outputs 4 times the number of input channels (e.g., 3 for RGB) to represent four directions (up, down, left, right), and we use a SoftPlus activation function to ensure non-negativity in the predicted rates. The hyperparameters are listed in Appendix H.

### 4.1 Image synthesis benchmarks

While Discrete Spatial Diffusion (DSD) was developed to enable generative modeling under a strict intensity constraint, we first evaluate its performance on conventional image synthesis benchmarks. Specifically, we trained DSD models on MNIST [45], CIFAR-10 [43], and CelebA [50], suggesting that DSD can approximate key statistical and structural properties of image distributions, despite operating under discrete-state and global conservation constraints.

We begin with MNIST, where intensity naturally corresponds to stroke thickness and digit area. In Fig. 3(a), we show inpainting results with a fixed mask with no-flux boundary conditions. During inference, particles inside the masked region rearrange to complete the digit structure, while the surrounding region remains fixed. Holding the visible structure constant, we varied the number of particles allowed to move, revealing that the total intensity governs which digit is most likely to emerge. This highlights the ability of DSD to incorporate hard constraints in downstream tasks such as inpainting. Additionally, we trained a conditional DSD model that employed a standard class-conditioning [69]. In Figure 3(b) we illustrate the class-conditioned generated images with different total numbers of particles, varying from low, typical, to high total intensities. While these

---

[3]Even though CFL condition is more commonly used in PDE integrators, the concept can be applied for our stochastic system. Suppose the reverse-time rate is $\bar{r}$. On average, the particle would move at a timescale $1/\bar{r}$ to one of its neighbors, traveling $\Delta x$. Then, the velocity $c = \hat{r}\Delta x$. The CFL condition is then $c\Delta t/\Delta x$ where in our scheme $\Delta t$ is the $\tau$; thus, the classical CFL convergence condition translates to the obvious bound of transition probability $\bar{r}\tau < 1$. This motivates us to ensure a conservative estimation of $\tau$, but enforcing a small $\bar{r}\tau$ to reduce the error.

do exhibit some artifacts, DSD notably learns the spatial structure of the digits and generates "Bolder" or "Lighter" digits without saturating the upper bound of the intensity (i.e. 255 for `uint8`). This would not have been precisely realizable using conventional diffusion models. Comparisons with other conditioning approaches can be found in Appendix E.

We also evaluated DSD on RGB datasets to explore scalability and generality. In Fig. 3(c,d), we show unconditional generations from models trained on CIFAR-10 and CelebA. Despite the discrete-state space and intensity-preserving dynamics, the model captures complex semantic structures from lighting, and textures to animals, vehicles, and human facial features. Appendix D includes large grids of generated samples, Fréchet Inception Distance (FID), and spatial FID (sFID) metrics, sampling ablations, and post-processing strategies for improving sample smoothness.

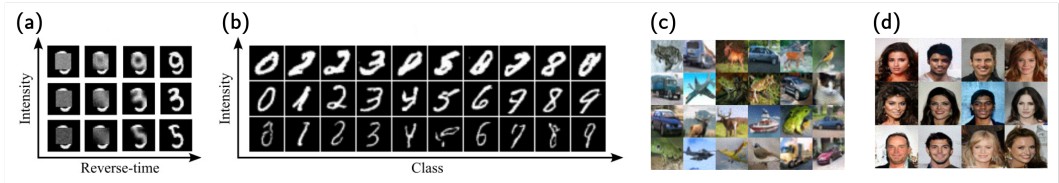

Figure 3: **(a)** Inpainting realizations on MNIST; 15% difference of conditioning intensity between consecutive rows. **(b)** Conditioned MNIST generations across different intensities and classes. **(c)** Unconditional CIFAR-10 generations. **(d)** Unconditional CelebA generations.

## 4.2 Subsurface rock microstructures

The microstructure of subsurface rocks governs a wide range of physical processes, including fluid transport, electrical resistivity, and mechanical deformation [7]. This originates from connected pores on the nano- and micro-scale, which vary in size, structure, and coordination degree across rock types. High-resolution 3D imaging via X-ray microtomography enables detailed pore-scale reconstructions, but these scans are expensive and limited to sample sizes on the order of millimeters to centimeters [19]. While direct imaging of rock microstructure is costly, measuring porosity (defined here as average intensity over the image) across large formations is inexpensive and can be performed without specialized equipment [48, 57]. This enables large-scale field measurements of porosity, even when high-resolution microstructural data is unavailable.

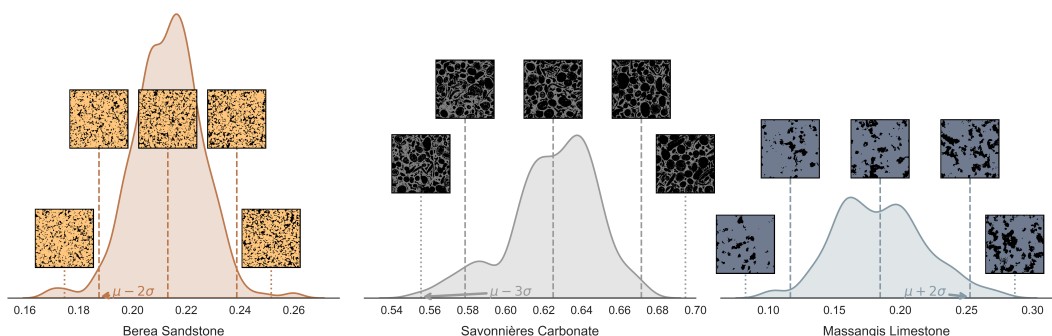

Figure 4: Distributions of porosity and generated samples for three rock classes. Each plot shows the porosity distribution across the dataset (shaded curve) and corresponding DSD-generated microstructures at selected quantiles. **(Left)** Berea Sandstone, **(center)** Savonnières Carbonate, and **(right)** Massangis Limestone. Samples illustrate the model's ability to generate structurally diverse images conditioned on exact global intensity. Quantitative metrics are presented in Appendix F.2.

To overcome this limitation, synthetic models are frequently used to generate representative pore structures for computational physics studies [56]. However, conventional reconstruction techniques impose strong geometric assumptions that fail to capture the heterogeneity observed in real rocks like Berea Sandstone, Savonnières and Massangis Carbonates. We trained DSD models using these rock samples, which represent a broad spectrum of pore structures (including granular, fossiliferous, and

dissolution-driven features) across two lithologies. A description of the training datasets is provided in Appendix F.1. Fig. 4 presents representative outputs from our models trained on 256×256 binary images. The generated samples accurately reproduce the key morphological and statistical features of the original datasets, including two-point spatial correlations and pore size distributions which are primary microstructural descriptors that govern permeability, tortuosity, and diffusion behavior in porous media (Fig. 19). Given that DSD allows for precise control over total porosity, one can generate synthetic microstructures that match the porosity measured in the field, enabling the reconstruction of representative pore-scale samples even in the absence of direct imaging.

### 4.3 Lithium-ion electrodes

Electrodes in lithium-ion batteries are porous materials with a complex microstructure that governs key properties like ion transport and electrochemical performance. Nickel-manganese-cobalt cathodes, among the most common, are composed of three phases: the active material driving the electrochemical reaction, the carbon binder ensuring electrical conductivity and mechanical stability, and the pore space filled with electrolytes.

The active material is expensive, creating a strong economic incentive to understand how its volume fraction and distribution influence electrode behavior. While tomographies are needed for studying microstructures and enabling computational modeling, acquiring diverse datasets is challenging [22]. To overcome this, researchers often rely on computational methods to generate synthetic microstructures [24]. While GANs have been explored for this purpose, they do not control phase volume ratio [29]. We trained a DSD model on tomography data [74], where two color channels were used to represent the carbon binder and active materials. The results, shown in Fig. 5, demonstrate DSD enables precise tuning of phase volume fractions. We then computed key morphological metrics characterizing the electrode structure [41], demonstrating the strong generative capability of the DSD method, which generalizes across the training set more effectively than existing approaches in the literature. For more details on datasets and reconstruction metrics applied to these samples, see Appendix G.

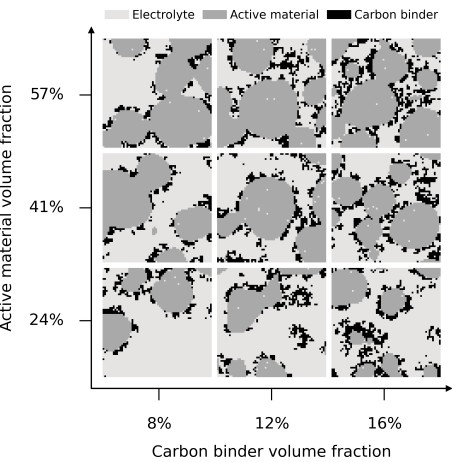

Figure 5: Generated cathode microstructures with varying phase volume fractions. The carbon binder domain appears in black, active material particles in gray, and electrolyte-filled pore space in white.

## 5 Limitations

The computational cost of forward sampling during training and reverse-time sampling during inference in DSD scales linearly with the *total intensity* of the image. While this makes DSD highly efficient for low-bit-depth or binary datasets, it may become less efficient than other techniques for higher-resolution images or datasets with higher intensity saturation, such as standard *uint8* images. While the FLOP cost of the forward noising process is higher than in conventional Gaussian diffusion models, this is handled entirely by CPU workers during the data loading phase, and so the added cost is effectively hidden completely behind the data pipeline and does not impact GPU throughput; DSD models were negligibly slower than their Gaussian counterparts, and models were trained in this work using one A100 GPU in $< 750$ hours per model (RGB images) and $< 50$ hours per model (microstructures, MNIST). Inference is slower (approximately $2\times$ per), due to the need to sample individual particle transitions, but affordable across datasets of practical interest. Despite this cost scaling, many of the largest available microstructural datasets in subsurface modeling are on the order of $1000 \times 1000$ resolution. We trained DSD on such large-scale scientific data, demonstrating that the method remains computationally feasible in this regime, also suggesting the feasibility of extending it

to 3D by carrying out implementation changes. Representative results from high-resolution generation are presented in Appendix F.3. Additionally, enforcing strict intensity conservation requires a custom forward process code (Eq. (1)) and a novel sampling scheme, deviating from conventional Gaussian diffusion models. This introduces a steeper learning curve for practitioners accustomed to standard diffusion approaches. However, we argue that these trade-offs are necessary to achieve exact intensity constraints, which is not possible with existing methods.

## 6   Conclusion

We introduced Discrete Spatial Diffusion (DSD), a fully discrete, intensity-preserving generative model approach for images and scientific data. The foundation is the use of discrete-state, continuous time statistical processes incorporating jump dynamics, rather than SDEs, and in particular is the first discrete diffusion model to explore spatially correlated corruption. DSD demonstrates competitive quality on standard benchmarks while enabling exact global constraints on particle count (thereby conserving image intensity or mass fractions under various applications) that are critical in many scientific applications. By preserving these constraints in both forward and reverse processes, DSD enables exactly constrained data generation, which we explored on image synthesis and domain-specific datasets. It also demonstrates that more complex statistical processes (in this case, random walks) can be used for diffusion modeling, opening the door for further models to exploit structure in their dynamics such as conservation laws and symmetries.

## Code availability

All code supporting this study is available at `https://github.com/lanl/DiscreteSpatialDiffusion`.

## Acknowledgments and Disclosure of Funding

This work was performed at Los Alamos National Laboratory (LANL), operated by Triad National Security, LLC, for the National Nuclear Security Administration of the U.S. Department of Energy (Contract No. 89233218CNA000001). The authors are supported by the Laboratory Directed Research & Development project "Diffusion Modeling with Physical Constraints for Scientific Data (20240074ER)".

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

## A  Deriving reverse-time transition rates

Here, we derive the reverse-time transition rate. Because the particles are moving independently, it is sufficient to discuss $n$ particles colocalized at $(x,y)$ in channel $c$, and the conclusion applies to other locations and color channels. For brevity, we will drop the $(x,y,c)$ dependence in this section when the context is clear. Let us index the particles by $i = 1 \ldots n = [I_t]_{x,y,c}$. For each of the $n$ particles, the reverse-time transition rates moving to $(x + \bar{\nu}_x, y + \bar{\nu}_y)$, where $(\bar{\nu}_x, \bar{\nu}_y) \in \{(-1,0),(1,0),(0,-1),(0,1)\}$ is

$$\bar{r}_{\bar{\nu}}^{[i]}(t) = r \frac{p_t\left(x + \bar{\nu}_x, y + \bar{\nu}_y, c | x_0^{[i]}, y_0^{[i]}, c_0^{[i]}\right)}{p_t\left(x, y, c | x_0^{[i]}, y_0^{[i]}, c_0^{[i]}\right)}, \tag{8}$$

according to the general theory of reverse-time dynamics for continuous-time Markov systems [11, 63]. We now perform the survival analysis for the inhomogeneous process. Within time $\mathrm{d}t$, the probability that particle $i$ leaves $(x,y,c)$ and moves to $(x + \bar{\nu}_x, y + \bar{\nu}_y, c)$ is $\bar{r}_{\bar{\nu}}^i(t)\,\mathrm{d}t + \mathcal{O}\left(\mathrm{d}t^2\right)$. As such, the probability of the particle remains at $(x,y,c)$ at time $t - \mathrm{d}t$ is $1 - \sum_{\bar{\nu}} \bar{r}_{\bar{\nu}}^{[i]}(t)\,\mathrm{d}t + \mathcal{O}\left(\mathrm{d}t^2\right)$. Thanks to the independence between the particle dynamics, the probability of *all* $n$ particles remaining at $(x,y,c)$ at time $t - \mathrm{d}t$ (recall that we are evolving the reverse-time dynamics) is $1 - \sum_{i=1}^n \sum_{\bar{\nu}} \bar{r}_{\bar{\nu}}^{[i]}(t)\,\mathrm{d}t + \mathcal{O}\left(\mathrm{d}t^2\right)$. Then, the probability of *no* particle leaving at a previous time $t - \Delta t$, where $\Delta t := N\mathrm{d}t$ is

$$\prod_{k=1}^N \left[1 - \sum_{i=1}^n \sum_{\bar{\nu}} \bar{r}_{\bar{\nu}}^{[i]}\left(t - (k-1)\,\mathrm{d}t\right)\mathrm{d}t\right] + \mathcal{O}\left(\mathrm{d}t^2\right), \tag{9}$$

which by sending $\mathrm{d}t \downarrow 0$ leads to the continuous-time survival function:

$$\mathbb{P}\{\mathcal{T} > t\} = \exp\left[-\int_0^t \sum_{i,\bar{\nu}} \bar{r}_{\bar{\nu}}^{[i]}(t')\mathrm{d}t'\right], \tag{10}$$

where $\mathcal{T}$ is the random time of the first particle moving out of $(x,y,c)$, the sum is over all possible directions and all particle index $i \in \{1 \ldots n\}$. Identifying the total rate $\sum_{i,\bar{\nu}} \bar{r}_{\bar{\nu}}^{[i]}(t')\mathrm{d}t'$ and the reverse-time transition rate for each particle and in each direction, Eq. (8), we arrived at Eq. (4).

## B  Training and generation algorithms.

Algorithm 1 gives the training algorithm using standard gradient descent techniques, and Algorithm 2 gives the inference algorithm used in this work.

---

**Algorithm 1** DSD training

---

Given the full transition probabilities $p_t(x', y', c'|x, y, c)$
**repeat**
    $I_0 \leftarrow$ a sample drawn from the training set
    Draw an index $k$ from $\{1, \ldots T\}$ uniformly
    **for** Each discrete intensity unit in $[I_0]_{x,y,c}$ **do**
        Draw $(x', y', c') \sim p_t(x', y', c'|x, y, c)$
        Move the unit from $(x, y, c)$ to $(x', y', c')$
    **end for**
    $I_{t_k} \leftarrow$ the corrupted image
    Compute the reverse transition rate Eq. (4)
    **if** Using $L^1$ rate-matching **then**
        Loss $\leftarrow \sum_{x,y,c,\bar{\nu}} \left| \bar{r}^{\text{NN}} - \bar{r} \right|$
    **else if** Using likelihood loss **then**
        Loss $\leftarrow -\log L$, defined in Eq. (7)
    **end if**
    Take a gradient step on $\nabla_\theta$Loss
**until** Converged

---

---

**Algorithm 2** DSD inference

---

Given CFL condition number $\varepsilon < 1$ and desired total intensities in the color channels, initiate an image $I_0$ with desired total intensities in the color channels
**for** Each discrete intensity unit in $[I_0]_{x,y,c}$ **do**
    Draw $(x', y', c') \sim p_1(x', y', c'|x, y, c)$
    Move the unit from $(x, y, c)$ to $(x', y', c')$
**end for**
$I_1 \leftarrow$ the fully corrupted image, $t \leftarrow 1$
**while** $t > 0$ **do**
    Evaluate NN predicted reverse rates $\bar{r}^{\text{NN}}_{\bar{\nu},x,y,c}$
    $\tau \leftarrow \min \left\{ t, \varepsilon \min_{\bar{\nu},x,y,c} \left( \bar{r}^{\text{NN}}_{\bar{\nu},x,y,c} \right)^{-1} \right\}$
    **for** each $(x, y, c)$ **do**
        Sample total moving particles:
        $n_\Sigma \sim \text{Binom} \left( [I_t]_{x,y,c}, \sum_{\bar{\nu}} \bar{r}^{\text{NN}}_{\bar{\nu},x,y,c} \right)$
        Sample a direction $\bar{\nu}$ for each moving particle:
        $n_{\bar{\nu}} \sim \text{Multinomial} \left( n_\Sigma, p_{\bar{\nu}} = \frac{\bar{r}^{\text{NN}}_{\bar{\nu},x,y,c}}{\sum_{\bar{\nu}'} \bar{r}^{\text{NN}}_{\bar{\nu}',x,y,c}} \right)$
        Move $n_{\bar{\nu}}$ intensity units to $(x + \bar{\nu}_x, y + \bar{\nu}_y)$
    **end for**
    Advance time: $t \leftarrow t - \tau$
    $I_t \leftarrow$ the configuration after movements
**end while**

---

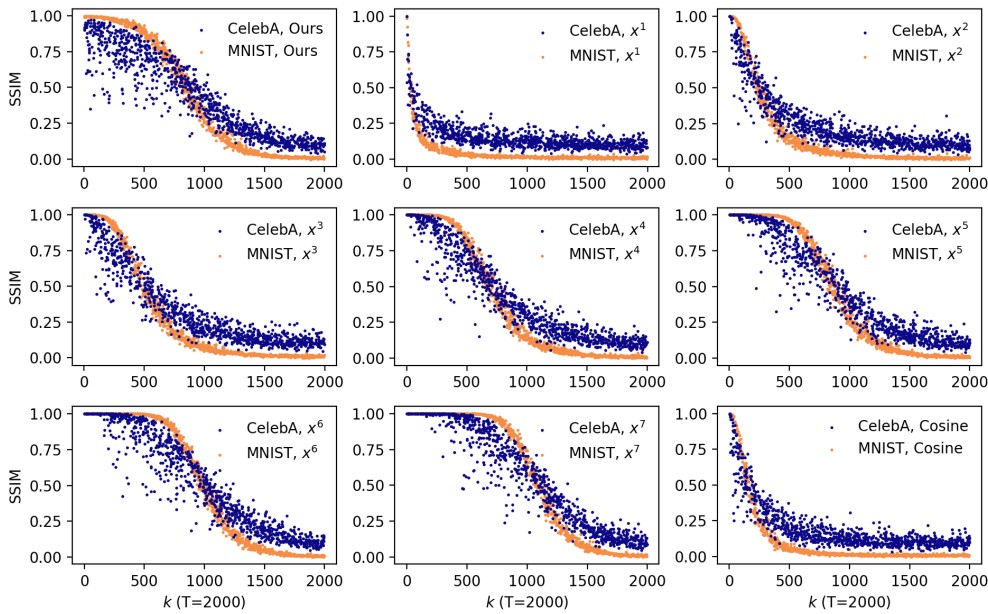

Figure 6: Structural Similarity Index Metric between the original and corrupted MNIST and CelebA images (1,000 samples evaluated at uniformly sampled random time $k \in \{1 \ldots 2000\}$) with various noise scheduler. We fixed $r = 120$ for MNIST and $r = 200$ for CelebA in this analysis. For ours, we use Eq. (2) with $\tau_1 = 7.5$ and $\tau_2 = 2.5$. For the polynomials, $t_k = (k/T)^n$, $n = 1 \ldots 7$. For the cosine schedule, we use the heuristic formula given in [55]. We remark that although the cosine schedule has been shown to be superior in previous studies [55, 62], the conclusion is based on the Ornstein–Uhlenbeck process, which is distinct from the spatial diffusion process (1).

## C  Spectral representation of DSD with periodic boundary conditions

Let us consider the two-dimensional DSD with periodic boundary conditions, whose master equation [77] is

$$\frac{\mathrm{d}}{\mathrm{d}t} p_{k,\ell}(t) = r p_{k-1,\ell}(t) + r p_{k+1,\ell}(t) + r p_{k,\ell-1}(t) + r p_{k,\ell+1}(t) - 4 r p_{k,\ell}(t). \tag{11}$$

Here, we use periodic boundary conditions, $p_{k\pm N,\ell} = p_{k,\ell}$, $p_{k,\ell\pm N} = p_{k,\ell}$ over the domain $(k,\ell) \in \{1, \ldots N_x\} \times \{1, \ldots N_y\}$. We can formally treat $p_{k,l}$ as a rank-2 tensor. We can write the one-dimensional discrete-space Laplace operator:

$$[\mathbf{L}_{x,y}]_{\mu,\nu} := \delta_{\mu+1,\nu} + \delta_{\mu-1,\nu} - 2\delta_{\mu,\nu} \tag{12}$$

where $\mathbf{L}_x$ is $N_x \times N_x$ and $\mathbf{L}_y$ is $N_y \times N_y$, $\delta$ is the Kronecker delta. Then, we can write the two-dimensional master equation succinctly as

$$\frac{\mathrm{d}}{\mathrm{d}t} \mathbf{p}(t) = r \left( \mathbf{L}_x \otimes I + I \otimes \mathbf{L}_y \right) \mathbf{p}(t). \tag{13}$$

We aim to identify the spectral representation of the full operator $L$. To achieve this, we first define the two-dimensional discrete Fourier and inverse Fourier transformations:

$$k_{m,n}(t) := \frac{1}{\sqrt{N}} \sum_{k=1}^{N_x} \sum_{\ell=1}^{N_y} e^{-2\pi i \left( \frac{mk}{N_x} + \frac{n\ell}{N_y} \right)} p_{k,\ell}(t), \tag{14a}$$

$$p_{k,\ell}(t) := \frac{1}{\sqrt{N}} \sum_{m=1}^{N_x} \sum_{n=1}^{N_y} e^{2\pi i \left( \frac{mk}{N_x} + \frac{n\ell}{N_y} \right)} k_{m,n}(t). \tag{14b}$$

To simplify the notation, we write $U_x$ and $U_y$ as the one-dimensional inverse discrete Fourier transformation

$$[\mathbf{U}_{x,y}]_{\mu,\nu} = \frac{1}{\sqrt{N}} e^{2\pi i \frac{\mu\nu}{N_{x,y}}}.$$ (15)

and again use the tensor notation. Then, Eqs. (14) can be succinctly represented as

$$\mathbf{k}(t) := (\mathbf{U}_x \otimes I)^\dagger (I \otimes \mathbf{U}_y)^\dagger \mathbf{p}(t),$$ (16a)
$$\mathbf{p}(t) := (\mathbf{U}_x \otimes I)(I \otimes \mathbf{U}_y)\mathbf{k}(t).$$ (16b)

The master equation in the spectral space can be derived now:

$$\begin{aligned}
\frac{\mathrm{d}}{\mathrm{d}t}\mathbf{k}(t) &:= (\mathbf{U}_x \otimes I)^\dagger (I \otimes \mathbf{U}_y)^\dagger \frac{\mathrm{d}}{\mathrm{d}t}\mathbf{p}(t) \\
&= r(\mathbf{U}_x \otimes I)^\dagger (I \otimes \mathbf{U}_y)^\dagger (\mathbf{L}_x \otimes I + I \otimes \mathbf{L}_y)\mathbf{p}(t) \\
&= r(\mathbf{U}_x \otimes I)^\dagger (I \otimes \mathbf{U}_y)^\dagger (\mathbf{L}_x \otimes I + I \otimes \mathbf{L}_y)(\mathbf{U}_x \otimes I)(I \otimes \mathbf{U}_y)\mathbf{k}(t) \\
&= r\left(\mathbf{U}_x^\dagger \mathbf{L}_x \mathbf{U}_x \otimes I + I \otimes \mathbf{U}_y^\dagger \mathbf{L}_y \mathbf{U}_y\right)\mathbf{k}(t)
\end{aligned}$$ (17)

because $\mathbf{U}_x^\dagger \mathbf{U}_x = \mathbf{U}_y^\dagger \mathbf{U}_y = \mathbf{I}$.

Let us now explicitly compute $\mathbf{U}_x^\dagger \mathbf{L}_x \mathbf{U}_x$:

$$\begin{aligned}
\left[\mathbf{U}_x^\dagger \mathbf{L}_x \mathbf{U}_x\right]_{k,\ell} &= \sum_{m=1}^{N_x} \sum_{n=1}^{N_x} [\mathbf{U}_x]_{k,m}^\dagger [\mathbf{L}_x]_{m,n} [\mathbf{U}_x]_{n,\ell} \\
&= \sum_{m=1}^{N_x} \sum_{n=1}^{N_x} \frac{1}{N} e^{2\pi i \frac{mk-n\ell}{N_x}} (\delta_{m+1,n} + \delta_{m-1,n} - 2\delta_{m,n}) \\
&= \sum_{m=1}^{N_x} \frac{1}{N}\left(e^{2\pi i \frac{mk-(m+1)\ell}{N_x}} + e^{2\pi i \frac{mk-(m-1)\ell}{N_x}} - 2e^{2\pi i \frac{mk-m\ell}{N_x}}\right) \\
&= \sum_{m=1}^{N_x} \frac{1}{N} e^{2\pi i \frac{m(k-\ell)}{N_x}}\left(e^{-\frac{2\pi i \ell}{N_x}} + e^{+\frac{2\pi i \ell}{N_x}} - 2\right) \\
&= \left(2\cos\left(\frac{2\pi\ell}{N_x}\right) - 2\right)\delta_{k,\ell} = 4\sin^2\left(\frac{2\pi\ell}{N_x}\right)\delta_{k,\ell}.
\end{aligned}$$ (18)

This shows that the process in the spectral space is diagonal with a cosine spectrum. The same analysis applies to $\mathbf{U}_y^\dagger \mathbf{L}_y \mathbf{U}_y$. Thus, element-wise, in the discrete Fourier space, we have

$$\frac{\mathrm{d}}{\mathrm{d}t}k_{m,n}(t) = 4r\left[\sin^2\left(\frac{\pi m}{N_x}\right) + \sin^2\left(\frac{\pi n}{N_y}\right)\right]k_{m,n}(t),$$ (19)

which means

$$k_{m,n}(t) = e^{4\left[\sin^2\left(\frac{\pi m}{N_x}\right) + \sin^2\left(\frac{\pi n}{N_y}\right)\right]rt}k_{m,n}(0).$$ (20)

In practice, we can always shift the particle of interest at $t = 0$ at the origin, i.e., $p_{0,0}(0) = 1$ and $p_{i,j} = 0$ if $(i,j) \neq (0,0)$. Then, $k_{m,n}(0) = 1/\sqrt{N_x N_y}$ and consequently

$$k_{m,n}(t) = \frac{1}{\sqrt{N_x N_y}} e^{4\left[\sin^2\left(\frac{\pi m}{N_x}\right) + \sin^2\left(\frac{\pi n}{N_y}\right)\right]rt},$$ (21)

so the analytical solution of $p_{k,\ell}$ can be expressed as

$$p_{k,\ell}(t) = \frac{1}{N_x N_y} \sum_{m=1}^{N_x} \sum_{n=1}^{N_y} e^{2\pi i\left(\frac{mk}{N_x} + \frac{n\ell}{N_y}\right) + 4\left[\sin^2\left(\frac{\pi m}{N_x}\right) + \sin^2\left(\frac{\pi n}{N_y}\right)\right]rt}.$$ (22)

Note that the double sum can be carried out in one two-dimensional Fast Fourier Transformation. The above equation allows us to compute the transition probability efficiently.

# D   Human-centric datasets

While our primary focus is on scientific data, we include results on standard vision benchmarks to contextualize DSD's performance to the computer vision community.

## D.1   MNIST experiments

As an initial validation, we trained an unconditional DSD model on MNIST and found that it reliably captures digit structure and diversity (Fig. 7).

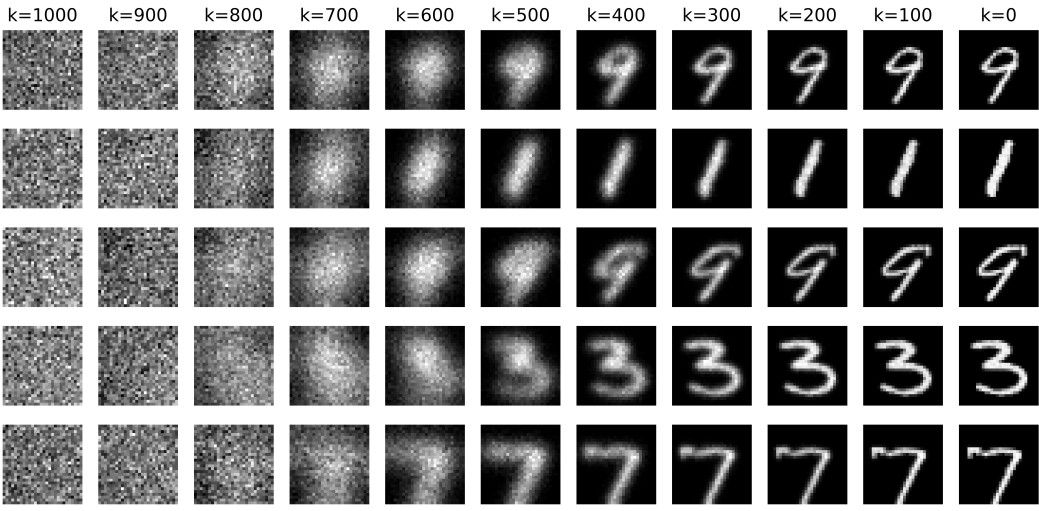

Figure 7: Unconditional MNIST generations.

Then, in our additional MNIST experiments, we explored class-conditional and inpainting generation. These experiments are particularly notable due to their interactions with the intensity-preserving property of DSD. For class-conditioning, we introduced class embeddings into our model following the approach described in [69]. Our model performed well at the task of class retrieval, consistently producing the desired class (Fig. 8).

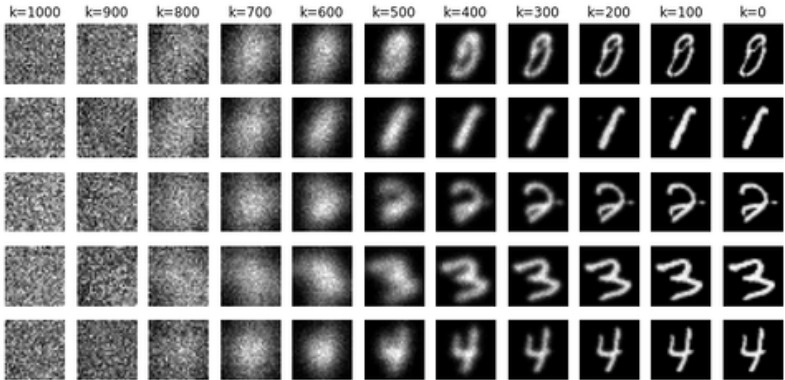

Figure 8: Class-conditional MNIST generations for digits 0 through 4. Each row corresponds to a specified target class.

For our intensity-related experiment, we tested our model on its ability to generate all of the classes given different starting intensity. Because generative models struggle to extrapolate beyond training data, our model demonstrated poor performance for certain digits on initial intensity values that were too high or too low. In response to this, we picked the '1' with the highest intensity for our high-intensity test, and the '0' with the lowest for our low intensity test, as 1 had the lowest intensity

of any of the numbers, and 0 had the highest. Our model performed very well on this task, consistently producing the target class even with varying intensity. See Fig. 3 (d) for results.

In training our model to perform inpainting, we shrunk the size of the transition matrix and held the rest of the image static. We obtained high quality generations very quickly, within 40K training steps. For our intensity-related experiment, we tested the model's reaction to increasing intensity within the inpainted region and were able to see different number generations from the same starting image (Fig. 9 ). See section E for comparisons with other conditioning approaches.

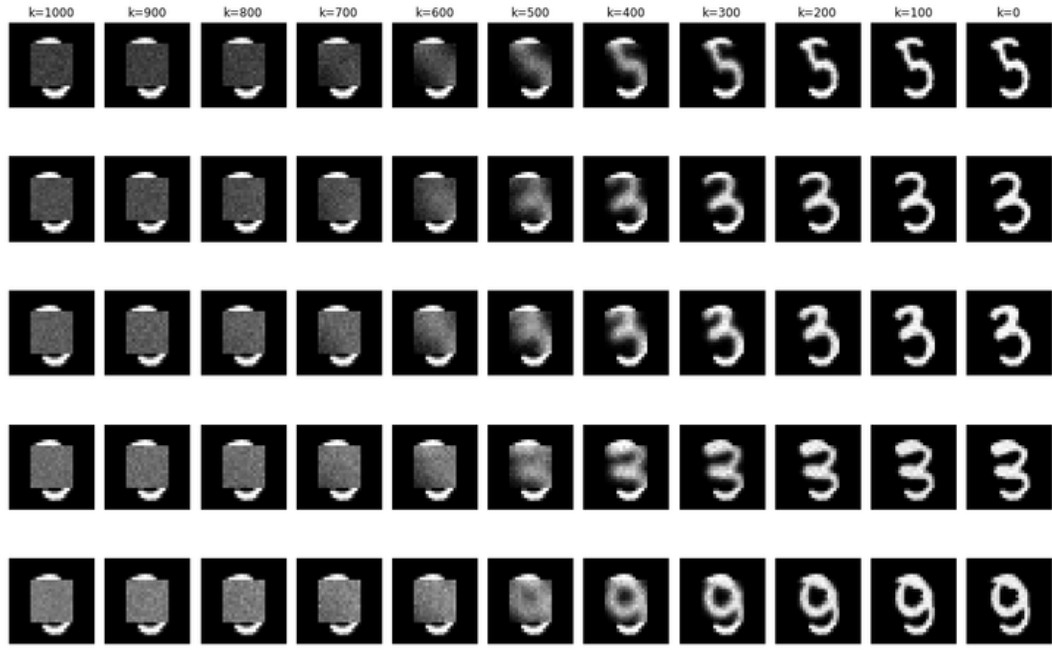

Figure 9: Unconditional MNIST inpainting with progressively increasing total intensity in the masked region. Rows correspond to increasing intensity levels, illustrating how digit identity and structure evolve under fixed spatial context.

## D.2 CIFAR-10 dataset

We trained DSD on the CIFAR-10 dataset and evaluated its generative performance under different sampling configurations. Specifically, we tested four values of the Courant–Friedrichs–Lewy (CFL) tolerance parameter, $\varepsilon \in 0.01, 0.05, 0.10, 0.15$, which controls the reverse-time integration step size. As expected, lower values of $\varepsilon$ yield more accurate reverse-time integration and lead to improved Fréchet Inception Distance (FID).

While FID is not the primary focus of our model, the discrete-state pixel-hopping nature of DSD introduces local high-frequency fluctuations during sampling (Fig. 10) that reduce smoothness. These artifacts, though minor, negatively impact FID and perceptual metrics. To address this, we apply a lightweight bilateral filter as a post-processing step to each generated image. This filter is non-trainable, respects both spatial proximity and intensity similarity, and acts as a smooth denoising operator that preserves semantic edges and attenuates visually incoherent fluctuations. Importantly, we apply it with default hyperparameters across all images, without per-sample tuning. The filtering operation is computationally negligible and can be implemented as a fixed convolutional layer at the end of the generation pipeline.

This simple step substantially improves both FID and spatial FID (sFID), without degrading the underlying structure of the samples (Fig.12). In Fig.11b, we quantify the impact of filtering across CFL tolerances. As an additional refinement step, we also use a pre-trained off-the-shelf CIFAR-10 classifier to discard semantically incoherent samples. Filtering based on classifier confidence can be integrated early in the sampling loop and correlates strongly with out-of-distribution or degenerate outputs (Fig. 11c).

A summary of results of these sampling experiments is shown in Table 1. While an FID score of around 20 does not represent state-of-the-art performance, we hypothesize that incorporating larger architectures, advanced optimization strategies, multi-GPU trainings, and data augmentation could further reduce this gap. DSD may be a viable foundation for discrete, constraint-preserving generative modeling even in conventional RGB settings.

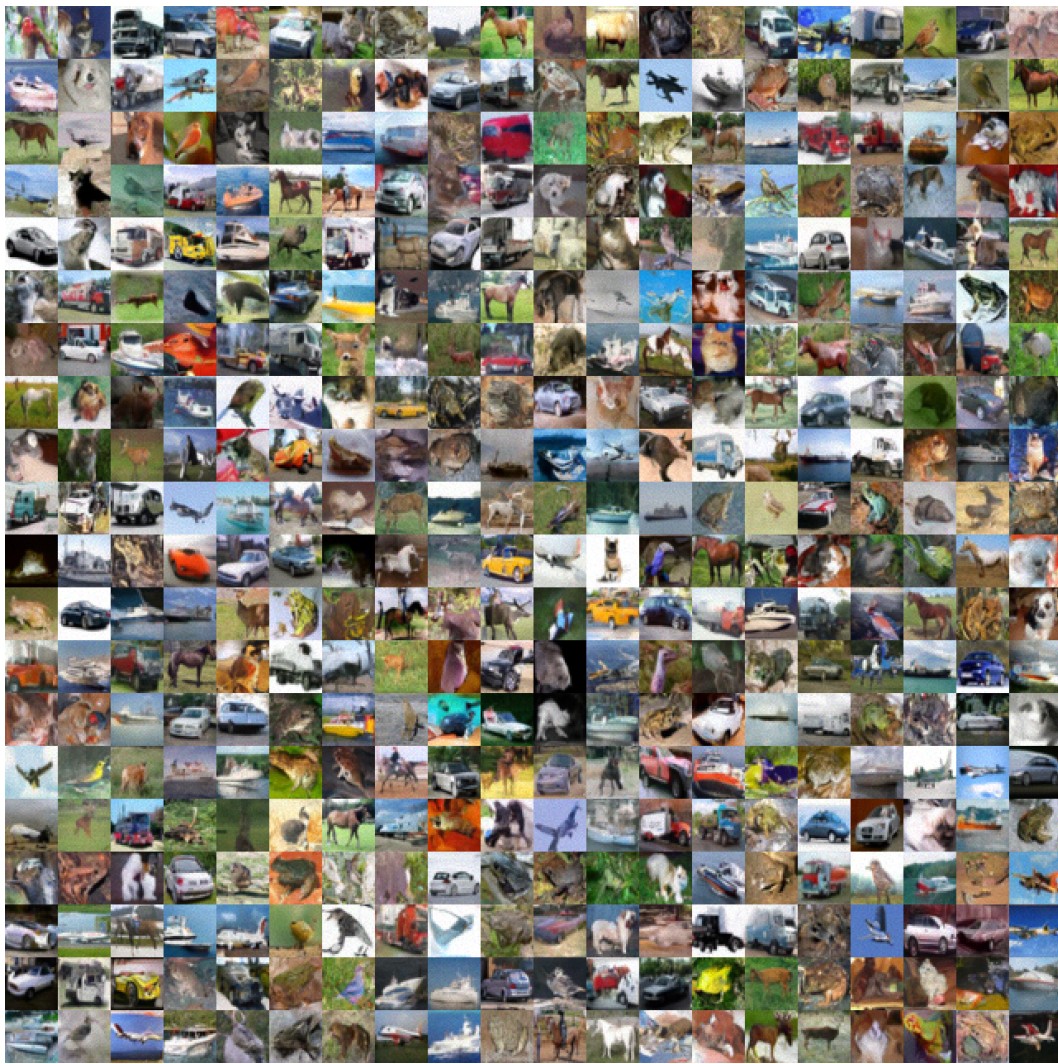

Figure 10: Unconditional CIFAR-10 generations produced by DSD with CFL tolerance $\varepsilon = 0.01$. Samples are generated without any post-processing. While structurally coherent, mild pixel-level irregularities are visible.

Table 1: Quantitative evaluation of CIFAR-10 generations under different sampling and filtering strategies (all scores computed on 50,000 images). The CFL tolerance $\varepsilon$ controls the reverse-time integration step size. Filtering denotes post-processing with a bilateral filter. Classifier thresholding retains samples with $p \geq 0.99$.

| Sampling Configuration | FID $\downarrow$ | sFID $\downarrow$ |
|---|---|---|
| CFL $\varepsilon = 0.01$ | 46.3 | 23.4 |
| CFL $\varepsilon = 0.01$ + filtering | 28.2 | 16.4 |
| + Classifier thresholding ($p \geq 0.99$) | 18.1 | 15.7 |

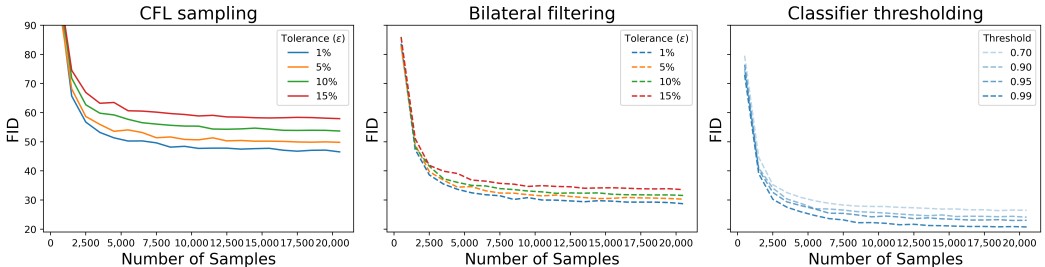

Figure 11: FID scores computed on 20,000 CIFAR-10 generations under different sampling configurations. **(Left:)** Samples generated with different CFL tolerances ($\varepsilon$). **(Middle:)** Same samples after applying a fixed bilateral filter. **(Right:)** Bilaterally filtered samples sampled with 1% further refined by discarding outputs with classifier confidence below a certain threshold.

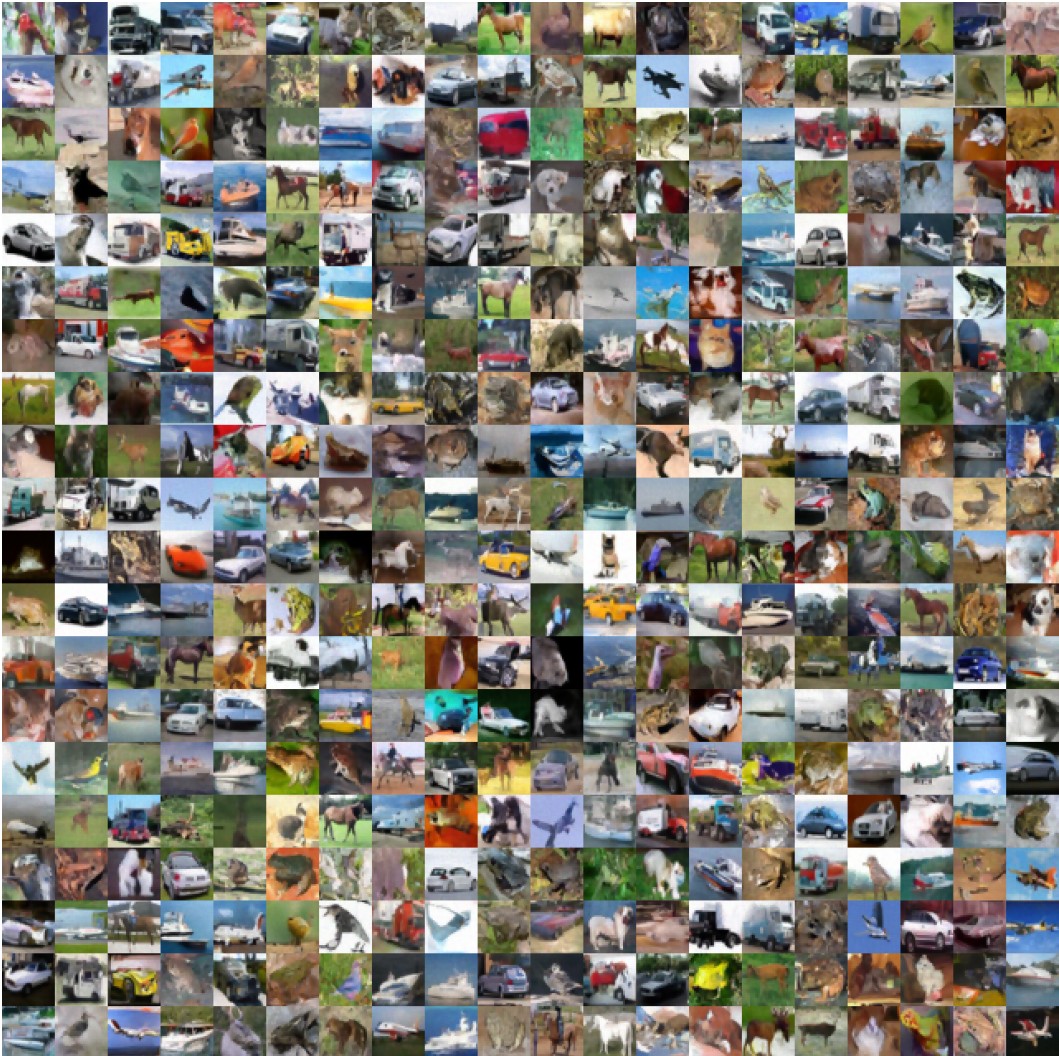

Figure 12: CIFAR-10 generations after bilateral filtering. These images correspond directly to the unfiltered samples shown in Fig. 10, post-processed using a fixed, non-trainable bilateral filter. The operation smooths pixel-level irregularities introduced by the discrete sampling process while preserving structural and semantic content.

### D.3 CelebA dataset

To demonstrate scalability to higher-resolution, human-centric data, we trained DSD on the CelebA 64×64 dataset. While the resulting FID of 29 is not competitive with state-of-the-art models, it remains consistent with our CIFAR-10 results and underscores the feasibility of applying DSD to RGB images with more complex structure. As illustrated in Figure 13, the model is capable of producing globally coherent facial samples, despite operating in a discrete, intensity-conserving setting. We speculate that such discrete spatial models could be beneficial for downstream applications involving controlled image manipulation, such as recolorization, region editing, or consistency-preserving transformations.

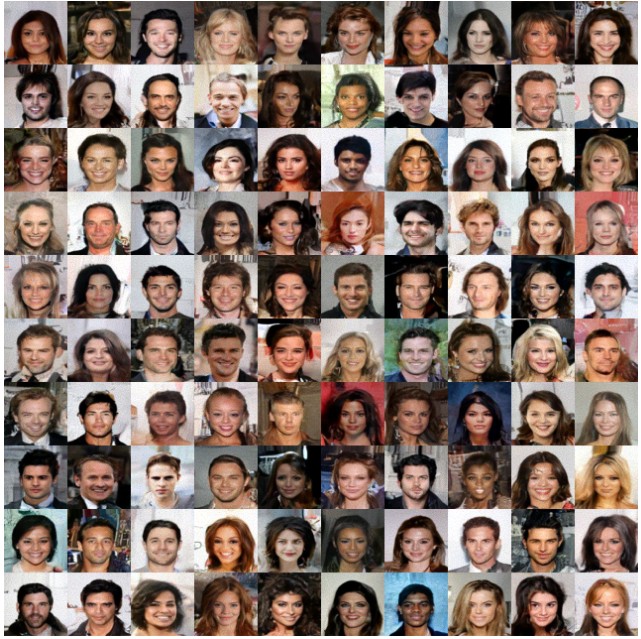

Figure 13: Unconditional generations of the CelebA 64×64 dataset.

Table 2: Quantitative evaluation of CelebA 64×64 generations under different sampling strategies. The CFL tolerance $\varepsilon$ controls the reverse-time integration step size. Filtering refers to post-processing with a bilateral filter. These scores were computed with 50,000 realizations.

| Sampling Configuration | FID ↓ | sFID ↓ |
|---|---|---|
| CFL $\varepsilon = 0.05$ | 44.2 | 29.0 |
| CFL $\varepsilon = 0.05$ + filtering | 28.9 | 25.7 |

## E Comparison with existing frameworks

### E.1 Gaussian Diffusion with Conditioning

To benchmark against conventional generative models, we implemented a Gaussian diffusion baseline conditioned on the target total intensity (or mass). The architecture employed a continuous-time variant of NCSN++ with explicit conditioning on total intensity per image via an auxiliary embedding channel. Fig. 14 shows the behavior of this model across a range of target intensities. Near the dataset mean, where conditioning remains within the empirical distribution, the model performs moderately well. However, closer examination reveals that the generated images often contain small negative values, even though training data was strictly bounded in $[0, 1]$. While this behavior may be inconsequential in floating-point representations, it becomes problematic when the outputs are discretized and clipped for evaluation or downstream use. In particular, thresholding negative values

and rounding to 8-bit precision systematically increases the intensity error. These issues become increasingly severe in the distribution tails, where the model frequently fails to respect the specified conditioning.

In contrast, our proposed DSD framework guarantees exact intensity preservation by construction, without requiring post hoc corrections, which holds uniformly across the intensity distribution, including extreme quantiles, as shown in Fig. 14. Notably, when sampling at the lowest intensity levels, DSD generates extremely thin or skeletal "1"s, while very high intensities produce over-saturated, bold structures such as "fat" zeros that span much of the image domain. This behavior suggests that the model is not merely enforcing conservation, but learning meaningful structural adaptations to accommodate different global constraints, consistent with underlying patterns in the data.

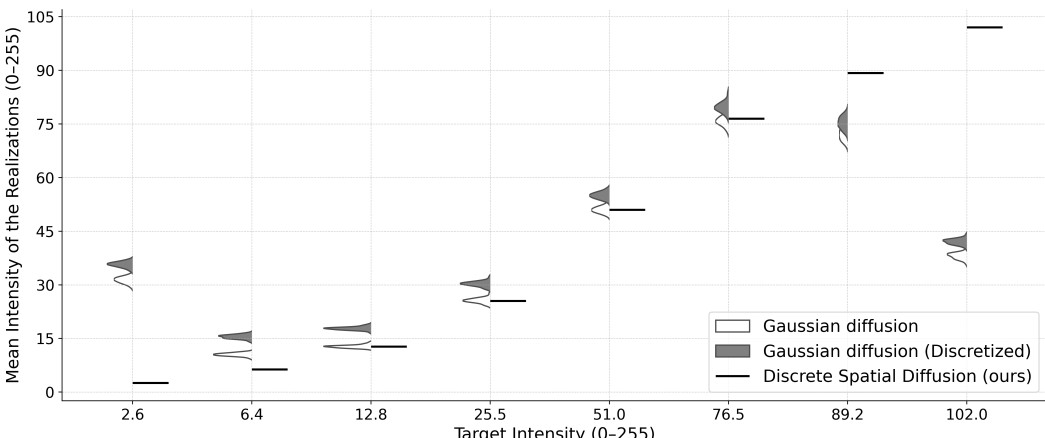

Figure 14: Sample-wise mean intensity across 1,000 realizations of mnist digits for each target intensity. We compare Gaussian diffusion **(white)**, its discretized version **(gray)**, and Discrete Spatial Diffusion **(black horizontal lines)**. Each violin shows the distribution of mean intensities computed across 1,000 individual samples.

Our observations indicate that: (1) intensity can be statistically conditioned within the distributional range, but without guarantee of exactness; (2) rounding and clipping of generated samples introduce further bias, revealing that the model relies on physically invalid negative intensities to approximate the constraint; and (3) extrapolation outside the data manifold results in complete failure of conditioning.

## E.2 Gaussian Diffusion with Conditioning for Scientific Data

To directly compare DSD with the standard continuous Gaussian diffusion framework in a scientific setting, we trained a continuous-time NCSN++ baseline with an explicit embedding of the target total intensity fraction (porosity) and evaluated it on the Savonniéres carbonate microstructure dataset (binary, 128×128). For each target we drew 100 samples, measured sample-wise porosity error, and inspected morphology. The Gaussian baseline tracks the target near the training range but degrades in the tails (large relative errors with low variance across samples) and exhibits characteristic artifacts—salt-and-pepper speckling and "staircase" grain boundaries consistent with conditioning that is only approximate and not enforced per-sample. These results are presented in Figure 15. DSD realizations withing the training distribution and well beyond the training range are shown in Figure 16 and Figure 17.

## E.3 Guidance Function Approach

In reinforcement learning settings, a guidance function can be naturally defined using task-specific reward or cost structures [40]. In generative modeling, however, such functions are generally unavailable, and often must be learned or heuristic in nature, as discussed in work on posterior sampling. Specifically, exact guidance is only possible when the conditional distribution as exact

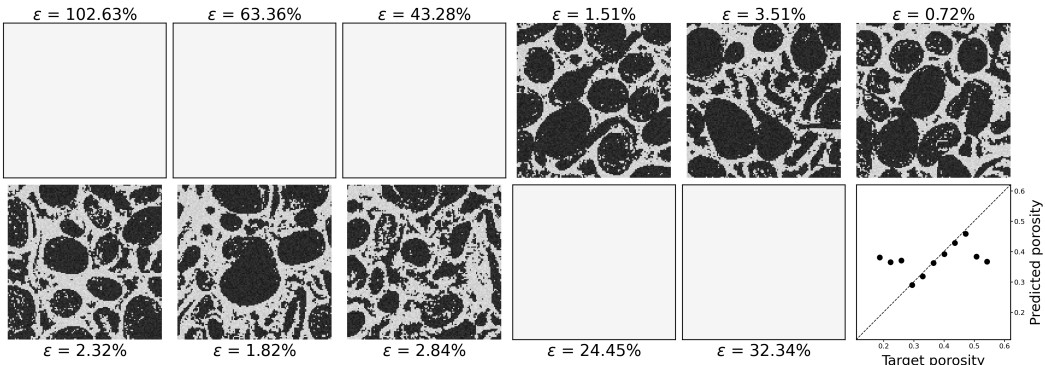

Figure 15: **Gaussian diffusion baseline with porosity conditioning on Savonnières carbonate.** (Binary, $128 \times 128$). The baseline approximately matches targets near the training regime but incurs large relative errors when extrapolating. Relative errors >20% are omitted from the plot.

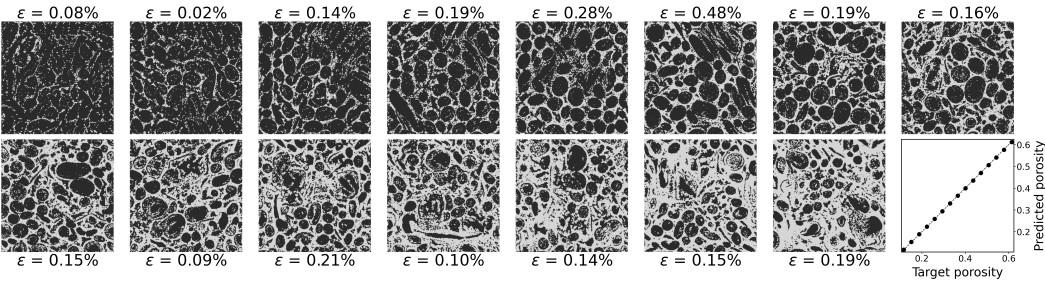

Figure 16: **DSD realizations of the Savonnières carbonate.** Samples generated with decreasing target porosity, extending well beyond the training range. Per-sample porosity closely follows the target (points lie on the identity line), indicating exact global intensity conservation by construction. Nevertheless, A negligible residual error $\varepsilon$ arises from a small fraction of overlapping pixels at the end of the reverse process. These appear as small white specks in the plots (visible in the online high-resolution version) and can be trivially corrected by a post-processing step that relocates overlapping pixels to the nearest empty voxel.

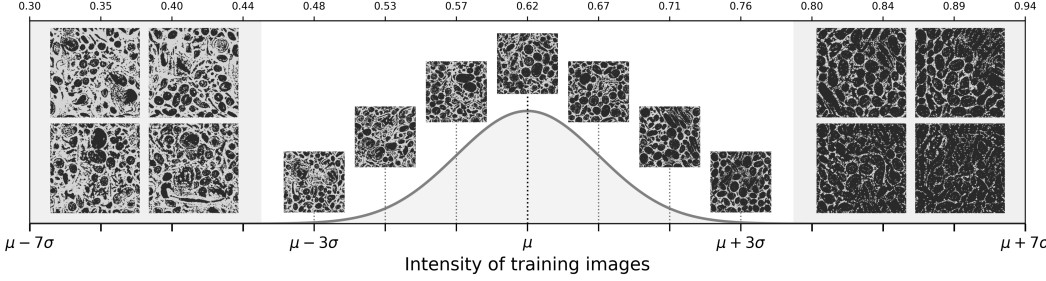

Figure 17: **Out of distribution DSD realizations of the Savonnières carbonate.** DSD-generated samples conditioned on total intensity from $\mu$-$7\sigma$ to $\mu$+$7\sigma$. Despite being trained near the data distribution center, the model produces coherent and physically plausible structures far beyond the training range, highlighting robust extrapolation enabled by exact intensity preservation.

posteriors (of generative samples given the constraint of interest) is known, which is not the case for total intensity constraints in image generation.

To explore whether heuristic guidance functions could serve as a surrogate, we conducted additional experiments applying a mean-squared error (MSE) penalty between the generated and target total intensity. Our results showed that while guidance can reduce error somewhat for intensities near the

mean training data intensity, it fails to enforce conditioning reliably, and especially so at the tails of the training distribution. In these cases, samples frequently severely violate the intensity constraint.

## F  Subsurface microstructures

### F.1  Detailed description of X-ray scans of subsurface rocks

- **Berea Sandstone**: This sandstone sample from [54] provides a high-resolution image of the rock microstructures obtained through X-ray microtomography (X-ray $\mu$CT). In this process, the rock sample is rotated while being scanned by an X-ray beam, capturing a series of 2D radiographs at different angles. These projections are then computationally reconstructed into a 3D volume, where each voxel represents the X-ray attenuation of the material at that location. The X-ray microtomography scans were performed using a SkyScan 1272 system, operating at 50 kV and 200 $\mu$A, with a CCD detector capturing projections at a resolution of 2.25 $\mu$m per voxel. The resulting dataset consists of grayscale images with a voxel size of 2.25 $\mu$m, where variations in intensity distinguish between the solid matrix and the pore space. The solid matrix primarily consists of tightly packed mineral grains—mostly quartz—while the pores are voids that can be occupied by fluids such as water or hydrocarbons. After preprocessing steps like contrast enhancement, noise reduction, and segmentation, the final dataset represents the pore network. The Berea sample has a measured porosity of 18.96% and permeability of 121 mD. This dataset is particularly useful for computational modeling, as it enables direct comparison between numerical simulations and experimentally measured permeability, providing a rich testbed for learning-based methods that seek to map complex microstructural information to macroscopic transport properties. This sedimentary rock is a well-characterized geological benchmark, widely used in fluid flow studies due to its homogeneous grain structure and consistent permeability properties, making it a good first benchmark for our study.

- **Savonnières Carbonate**: This carbonate sample, described in [9], is a layered, oolithic grainstone with a wide porosity and a permeability varying from 115 to over 2000 mD, depending on local heterogeneities. The rock is characterized by a highly multimodal and interconnected pore structure, with distinct macropores and microporosity. X-ray microtomography (X-ray $\mu$CT) was used to image the sample at a resolution of 3.8 $\mu$m voxel size, revealing intricate pore geometries. The sample was scanned at the Ghent University Centre for X-ray Tomography (UGCT) using their HECTOR scanner, developed in collaboration with XRE, Belgium. The macropores include both intergranular voids and hollow ooids, while the microporosity is found within ooid shells and intergranular spaces. Micropores in the sample often serve as the primary pathways connecting poorly connected macropores, creating a complex hierarchical network. After preprocessing steps, including noise reduction, anisotropic diffusion filtering, and watershed segmentation, a multiscale pore network model was extracted. This dataset is particularly compelling due to its extreme heterogeneity, with pore sizes spanning orders of magnitude, and its ability to represent coupled serial and parallel flow pathways. Savonnières serves as a test case for studying the impact of complex samples in our workflow.

- **Massangis Limestone**: This oolitic limestone sample from [8] is a highly heterogeneous carbonate rock with a complex, multimodal pore structure resulting from diagenetic alterations, including dolomitization and dedolomitization. The rock contains a mix of intergranular and moldic macroporosity, along with microporosity concentrated in ooid rims and partially dissolved dolomite regions. Its porosity ranges from 9.5% to 13.8%, depending on local variations, and its permeability is highly anisotropic due to the interplay between connected macropores and poorly accessible microporosity. X-ray microtomography (X-ray $\mu$CT) was used to image the sample at a voxel resolution of 4.54 $\mu$m, capturing the intricate connectivity of macro- and micropores. The sample was scanned at the Ghent University Centre for X-ray Tomography (UGCT) using a FeinFocus FXE160.51 transmission tube, in collaboration with Paul Scherrer Institute (PSI), Switzerland. Differential imaging was applied to enhance the detection of fluid-filled microporosity, revealing the rock's internal heterogeneities. Unlike more uniform carbonate samples, Massangis exhibits significant spatial variations in pore connectivity, leading to zones of high permeability interspersed

with isolated pore networks. This dataset serves as another challenging benchmark for modeling porous media microstructure.

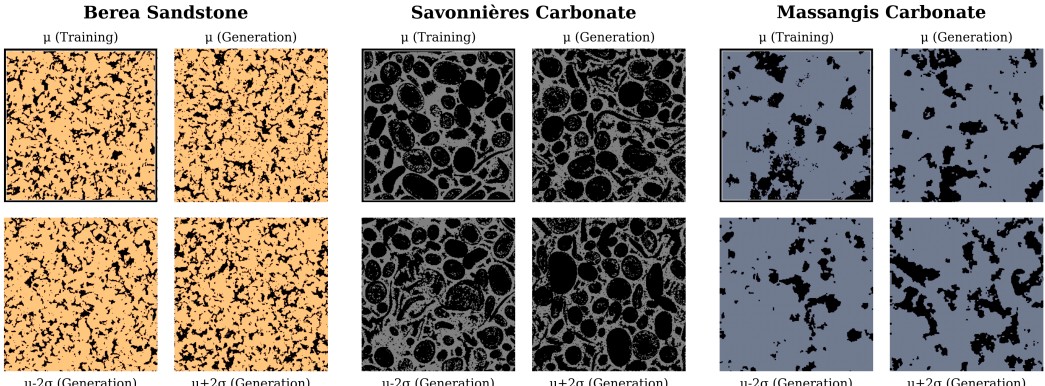

Figure 18: Schematic representation of three rock types: Berea Sandstone, Savonnières Carbonate, and Massangis Carbonate. The first image **(top left)** for each rock type shows one training sample, while the second one **(top right)** displays the generated sample conditioned on the mean intensity $\mu$ of the training set. The third **(bottom left)** and fourth **(bottom right)** samples illustrate the generated samples conditioned on $\mu - 2\sigma$ and $\mu + 2\sigma$, respectively, where $\sigma$ represents the standard deviation of the training set intensity distribution.

## F.2 Quantitative metrics of the generated images

In porous media analysis, characterizing the spatial arrangement and size distribution of pores is crucial for understanding transport properties, mechanical behavior, and overall structure-function relationships. To quantify these characteristics, we compute the spatial correlation function and pore size distribution (PSD) using PoreSpy [34], a Python-based toolkit for quantitative analysis of porous media images. The Pore Size Distribution (PSD) characterizes the variation of pore sizes within a porous material, providing insights into connectivity, permeability, and flow dynamics. The most common method to determine PSD computationally is the local thickness approach. Given a binary image $I(x, y)$, where pore space is represented as 1 and solid space as 0, the pore size function $f(r)$ is defined as the probability density function (PDF) of the largest sphere that can be inscribed at any point within the pore space. The PSD provides a statistical summary of pore connectivity and transport properties. Small pores dominate permeability, while large pores govern bulk flow. Both of these metrics for the training and generated samples are shown in Fig. 19, our realizations exhibit **excellent** agreement with the training data.

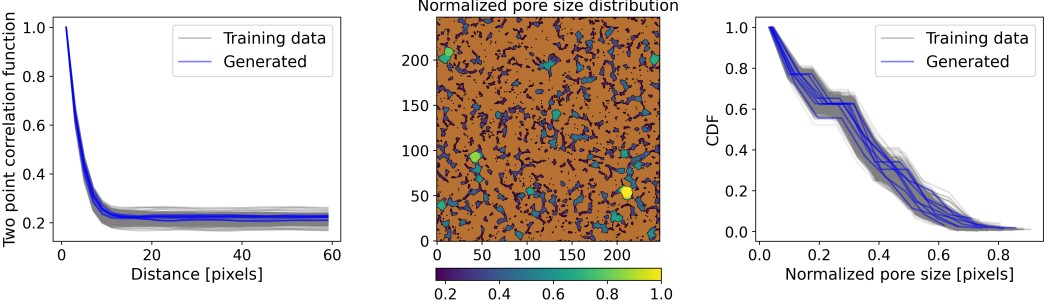

Figure 19: Quantitative comparison between training and generated rock samples. **(Left)** Two-point correlation function, showing excellent agreement of spatial features between training data (gray) and 100 randomly generated samples (blue). **(Middle)** Normalized pore size distribution schematic, with colors indicating relative pore sizes. **(Right)** Cumulative distribution function (CDF) of the normalized pore sizes, comparing the statistical distribution of training and generated samples.

Additionally, following the evaluation protocol in Lee and Yun [46], we computed Fréchet Inception Distance (FID) for our generated carbonate samples, obtaining a score of 0.9, substantially lower than their reported value of 18.1 on rocks of similar lithology. While this comparison is not strictly one-to-one due to dataset and preprocessing differences, it offers a compelling quantitative indication of the improved realism and fidelity of DSD-generated microstructures. We note, however, that FID may have limited interpretability for binary porous media images, and should be interpreted alongside domain-specific metrics.

### F.3 Large scale generations.

To evaluate the scalability of DSD in realistic scientific settings, we trained a model on high-resolution binary microstructure data of Leopard Sandstone, an interesting sample from the geological standpoint. This dataset consists of $1000 \times 1000$ samples representing complex subsurface structures characterized by heterogeneity, anisotropy, and nontrivial pore connectivity. Generating such large samples is particularly relevant for studying representative elementary volumes.

Despite the increased resolution and number of particles, DSD remains computationally feasible in this regime. The forward process remains efficiently parallelizable across CPU workers, and reverse-time sampling scales linearly with intensity while maintaining tractable runtimes. A subset of generated samples is shown in Fig. 20, demonstrating the model's ability to reproduce structurally diverse and geologically plausible microstructures at scale.

While extending DSD to 3D requires modest modifications to the implementation (e.g., defining a 6-connected lattice for particle jumps), we find the approach remains tractable for volumetric data, further supporting its suitability for scientific applications in porous media and materials modeling.

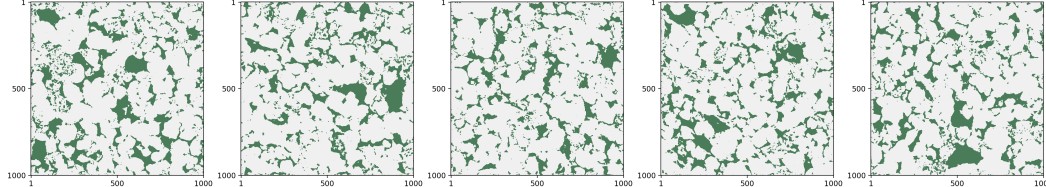

Figure 20: Realizations of Discrete Spatial Diffusion (DSD) on $1,000 \times 1,000$ images of Leopard Sandstone. These samples demonstrate that our model scales to high-resolution, structurally complex porous media. Leopard Sandstone exhibits heterogeneous, anisotropic pore structures representative of real-world geological variability.

## G X-ray scans of NMC cathodes

### G.1 Dataset description

This dataset provides high-resolution 3D images of a Li-ion battery cathode composed of active material (nickel-manganese-cobalt oxide, NMC), carbon black, and a polymer binder [74]. The cathode sample was imaged via X-ray microtomography (X-ray μCT) and nano-tomography (X-ray nano-CT) to capture both the overall electrode architecture and fine-scale features of the carbon/binder domain (CBD). For micro-CT, a Zeiss Xradia Versa 520 system was operated at 80 kV and 88 μA, acquiring projections at an effective isotropic voxel size of approximately 398 nm over a field of view of about 400 μm. The nano-CT scans were performed using a Zeiss Xradia Ultra 810 system with a chromium target (35 kV, 25 mA), yielding isotropic voxel sizes on the order of 126 nm across a field of view of approximately 64 μm. In both cases, the 2D radiographs were reconstructed into 3D grayscale volumes using a filtered back-projection algorithm, capturing the X-ray attenuation due to the dense NMC particles and the less attenuating pore/CBD regions.

These tomographic datasets reveal the hierarchical microstructure of the electrode, from tens-of-micrometers NMC active particles to nanometer-scale pores within the percolated carbon network. After preprocessing—such as non-local mean filtering, contrast enhancement, and slice-by-slice local thresholding—segmentation identifies three main phases: (1) the NMC active material, (2) the CBD, and (3) the pore space. Measured porosity values for these cathodes can exceed 30%, while the typical

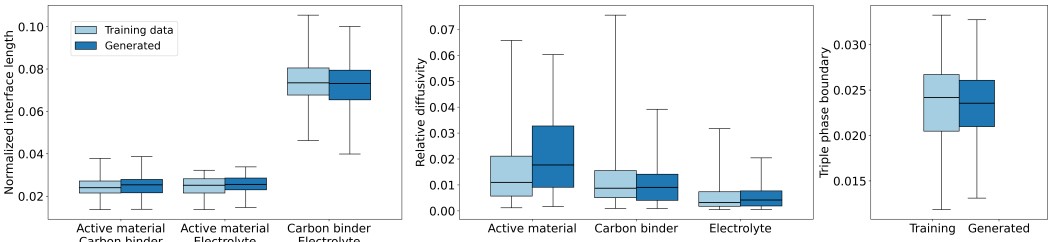

Figure 21: Microstructural characterization metrics for 80 training samples and 80 generated samples. The boxes show the 25th-50th-75th percentile, the whiskers the minimum and maximum values. Metrics computed using TauFactor [41].

volume fraction of active material is on the order of 40%. The overall areal loading of the active material is around 29.78 mg·cm$^{-2}$, corresponding to about 33 mAh·cm$^{-2}$ in specific capacity. These 3D reconstructions enable computational modeling of transport properties (e.g., tortuosity factor) and electrochemical performance, facilitating direct comparisons with experimentally measured parameters. Because of the electrode's well-defined spherical NMC particles and percolating carbon network, this dataset serves as a robust benchmark for multi-scale modeling and data-driven methods that aim to link microstructural features to macroscopic cell behavior.

## G.2  Effective metrics

The analysis of NMC cathode tomography and the generated images was conducted using three metrics: interface length, triple-phase boundary, and relative diffusivity. These metrics are essential for quantifying the morphological and transport characteristics that influence the electrode's electrochemical performance. Below we describe these metrics in detail.

**Interface length** refers to the total length of boundaries where two distinct phases, such as active material and pore or electrolyte, intersect. A higher interface length indicates more active sites for electrochemical reactions and enhances ion transport pathways, thereby improving the electrode's overall performance. This metric is calculated by identifying and summing the perimeters of all phase boundaries in the segmented image.

**Triple-Phase Boundary** denotes the regions where three different phases—typically solid active material, electrolyte, and a conductive phase or pore space—converge in the microstructure. TPBs are crucial for facilitating efficient electrochemical reactions, as they provide optimal sites where all necessary phases interact. The total TPB length is determined by locating points or lines where three phases meet and summing their lengths within the image.

**Relative Diffusivity** quantifies the reduction in ion transport within the porous cathode structure relative to an unobstructed medium. It is defined as the ratio of the effective diffusivity, $D_{\text{eff}}$, through the porous medium to the intrinsic diffusivity, $D_0$, of the conductive phase: $D_{\text{rel}} = D_{\text{eff}}/D_0$. This reduction is primarily attributed to the geometric complexities of the microstructure, encapsulated by the tortuosity factor, $\tau$, in fact $D_{\text{rel}} = D_{\text{eff}}/D_0 = V_f/\tau$, where $V_f$ is the volume fraction of the phase under analysis.

We computed these metrics using the Python library TauFactor [41], and the comparisons between the real and generated images based on these metrics are illustrated in Fig. 22, while a collection of the training data and generated images is in Fig.21.

# H  Hyperparameters for experiments

In our experiments, we thoroughly tested our model on various hyperparameters using the MNIST dataset. The MNIST dataset was chosen as a baseline for hyperparameter testing due to its low computational training cost. We found that our model was very robust with respect to the hyperparameters used, consistently generating quality generations without hyperparameter tuning. Due to limited compute, only limited tests were performed on CelebA, but we hypothesize that our model would perform well with different hyperparameters than the ones used. For the choice of our 'r', we chose a

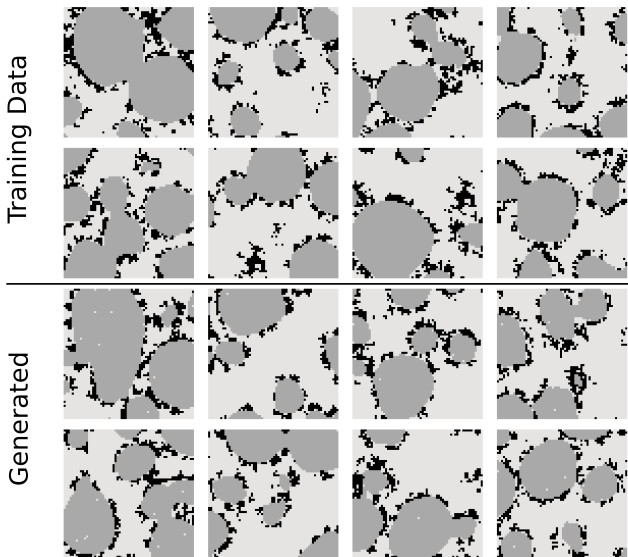

Figure 22: **(Top)** Eight randomly picked samples from the NMC cathodes dataset. **(Bottom)** Random unconditional realizations of our model.

rate that was large enough to allow full degradation, enabling the model to learn to predict starting from full noise. See 3 for our hyperparameters used.

Table 3: Hyperparameters used in all our experiments. All models were run in a single NVIDIA A100 (or similar).

| Dataset | $r$ | Schedule | Boundary Condition | Loss | CFL Tolerance | Channel Multiplier | Training Iterations | Notes |
|---|---|---|---|---|---|---|---|---|
| MNIST | 120 | Ours | Periodic | Eq. 5 | 0.15 | (2,2,2) | 100K | unconditional |
| MNIST | 120 | Ours | Periodic | Eq. 6 | 0.15 | (2,2,2) | 90K | unconditional |
| MNIST | 120 | Ours | No-flux | Eq. 6 | 0.15 | (2,2,2) | 80K | unconditional |
| MNIST | 120 | Ours | No-flux | Eq. 6 | 0.11 | (2,2,2) | 70K | class-conditioned |
| MNIST | 85 | Ours | No-flux | Eq. 6 | 0.07 | (2,2,2) | 40K | inpainting (14x14) |
| CIFAR10 | 160 | Ours | Periodic | Eq. 6 | Fig. 11 | (1,2,2,2) | 10M | unconditional |
| CelebA | 200 | Ours | No-flux | Eq. 5 | 0.05 | (1,2,2,2) | 3M | unconditional |
| Electrodes | 200 | $x^5$ | Periodic | Eq. 6 | 0.01 | (1,2,2,2) | 180k | |
| Rocks | 250 | $x^4$ | Periodic | Eq. 6 | 0.1/0.2/0.05 | (1,2,2,2) | 50k | tolerance avoids overlapping particles |

