# OpenReview forum: "Discrete Spatial Diffusion: Intensity-Preserving Diffusion Modeling"
_NeurIPS.cc/2025/Conference — NeurIPS 2025 spotlight_

### Official Review · Reviewer_RTj5 · 2025-06-11

**Clarity:** 3
**Significance:** 3
**Originality:** 3
**Rating:** 5
**Confidence:** 3

**Summary:**

The article is about generating artificial images (or other data) by reversing a random process with a neural network. Image generation with a Gaussian diffusion reversing neural network is a popular topic. In this process, an image is generated from Gaussian noise by iteratively reversing the diffusion. In this process, the pixel values of the images are supposed to be continuous.

The article proposes a new random process in which the image (or data) has discrete values, with the constraint that the total mass (sum of all pixel values) is preserved between each random step in each color channel. This process can be applied to RGB images because they are originally uint8, but it is particularly suited for generating other types of data, such as mineral tomography. This data is binary, and controlling the amount of porosity (number of zeros) and the shape of the cavities is important so that they look natural. Thus, porosity can be controlled at initialization by the number of zeros in the initial random image.

The article presents the forward discrete mass-preserving process in detail. This process can be represented in two ways: with the random positions of individual particles or with population histograms of each cell in the image. In the forward process, each particle has an equal probability of jumping into one of its four neighboring cells or of not moving. The backward process is similar but different. Because it is conditioned by the initial position of the particle, the directions are no longer equiprobable, as the particle will be pushed to return to its initial state.

As with Gaussian diffusion, the discrete process is reversed using a network. The network is trained to estimate the probability that a particle will reverse jump in each direction. The authors propose two loss functions: one that minimizes the L1 distance between the network guess and the true backward process and one that maximizes the likelihood of the network in the backward process.

In the "Experiment" section, the authors demonstrate that their process can generate RGB images similar to those produced by Gaussian diffusion. Then, they present results on tomography generation, demonstrating that their method enables them to control porosity while producing realistic data.

**Questions:**

(0) My main concern is that the results of classical Gaussian diffusion should be provided for comparison in experiments. Even if the results are similar, this new framework will still be interesting.

(1) Regarding my criticism of the notation, I think it would be clearer if the forward equations (1a, b) and backward equation (4) used probabilistic notation and definitions. For example, I think it would be better to describe the  "noise" $\nu$ as categorical random variables with:
\begin{equation}
\mathbb{P}(\nu) = \begin{cases}
r & \text{if } \nu\in \{(0,1),(0,-1),(1,0),(-1,0)\};\\
1-4r & \text{if } \nu = (0,0); \\
0     & \text{otherwise}
\end{cases}
\end{equation}

It may help to understand the likelihood in equation (6). Personally, I have trouble seeing where it comes from.

(2) Table 3 should be cited in the text of the experiments so that readers know the experimental setup is provided in the supplementary material.

(3) Equations (5), (6), and (7) should state the network inputs and variables on which the sums and integrals are based to make them easier to understand. For example, in equations (6) and (7), it is unclear what the sums are.

(4) line 657, I think there is a typo and that $p_{i,l}$ should be replaced with $p_{k,l}$.

**Ethical Concerns:**

["NO or VERY MINOR ethics concerns only"]

**Final Justification:**

The article proposes a new generative method based on an original stochastic process that allows for mass conservation. This is both original and useful in certain application scenarios. The article is detailed and well explained.

I just thought that, in order to demonstrate the usefulness of the method, the article needed to draw a comparison with classical Gaussian diffusion for these specific applications. The authors carried out the experiments and made the small corrections to the text that I requested. I also raised the concern that the mathematical notations were not standard, but the authors showed me that I was mistaken.

I no longer have any concerns about this article. I think it's good work that should be published, which is why I changed my rating to Accept.

**Limitations:**

yes

**Paper Formatting Concerns:**

No formatting issues

**Quality:**

3

**Strengths And Weaknesses:**

Strengths :

The article proposes an interesting alternative to Gaussian diffusion. The random forward and backward processes are explained quite well. The authors demonstrate that this process works on natural RGB images, even though that is not their main target application. They also acknowledge the limitations, explaining that this process is not efficient for RGB images because the number of random variables is higher than in Gaussian diffusion.

The article is well-written and covers all aspects of this type of solution (random process, forward-backward, loss, sheduler ..), with extensive citation of applicative and theoretical articles.

The article also provides extensive supplementary material, including all the necessary mathematical derivations and additional experiments.

In this sense, I find that the article provides a thorough description of the proposed method.

Weaknesses :

Even though the article proposes many interesting experiments, I think an important one is missing. The article claims that Gaussian diffusion is not suited for tomography generation and that the proposed process is better. Therefore, I think the experiment should highlight this fact by comparing the proposed solution with tomographies generated using the Gaussian process  ( with a conditioning on porosity and relaxation from discrete/binary to continuous).

In my opinion, since the article describes a random process, it should use more common probabilistic notations and definitions. This would make it easier to compare to other methods of this kind.

---

> ### Author Rebuttal · Authors · 2025-07-30
>
> Thank you for the constructive feedback and for recognizing the novelty and thoroughness of our method. We believe the new experimental results and clarifications further strengthen the case for this work’s significance, particularly in scientific domains where conservation laws are important.
>
> **[Q0]**
>
> Thank you for encouraging us to include a comparison with a Gaussian diffusion framework. This was a valuable suggestion, and we have carried out the experiment accordingly. We conducted the experiment using our Savonnières Carbonate sample, which displays the widest range of heterogeneous geological attributes in our dataset, including  dissolution features, fossils, and grain inclusions. The results confirm key limitations of Gaussian diffusion for our target application. We will document this experiment thoroughly in the final version, although we cannot include sample visualizations here, we will do our best to describe the quantitative and qualitative results:
>
>
> - The generated samples are noticeably “grainy” and exhibit artifacts.
>
>
> - Grain boundaries are not perfectly smooth or round (compared to our samples in Figure 4 and 15) but appear as staircase-like structures (like those in Minecraft).
>
>
> - Salt-and-pepper noise is present both inside and outside the solid regions.
>
>
> - While the porosity (i.e., the mean pixel intensity over image size) is approximately correct (especially near the mean ~0.6), this is achieved through a bimodal distribution centered near 0 and 1, with values falling outside the [0,1] range (e.g., down to –0.2 and up to 1.2). As a result, binarization does not significantly worsen the porosity error, unlike with datasets like MNIST (Figure 14).
>
> - We trained the model using data whose porosity is between 53% and 72%. We deployed the model to a wider range of target porosity between 45.74% and 81.24%. The model confidently produces incorrect predictions (low standard deviation over 100 generated samples) outside of the training distribution, indicating poor generalization outside the center of the distribution. In contrast, DSD was able to generalize, since intensity is a hard constraint in DSD.
>
>
>
>
>
>
>
>
>
>
> | **Porosity \[%]** | **Prediction \[Mean ± Std]** | **Relative Error \[%]** |
> | ----------------- | ---------------------------- | ----------------------- |
> | 45.74             | 62.26 ± 0.60                 | 36.11                   |
> | 49.29             | 61.94 ± 0.45                 | 25.67                   |
> | 52.84             | 54.85 ± 0.54                 | 3.80                    |
> | 56.39             | 57.08 ± 0.52                 | 1.22                    |
> | 59.94             | 60.33 ± 0.49                 | 0.66                    |
> | 63.49             | 63.88 ± 0.53                 | 0.61                    |
> | 67.04             | 67.61 ± 0.48                 | 0.84                    |
> | 70.59             | 71.30 ± 0.55                 | 1.00                    |
> | 74.14             | 62.87 ± 0.45                 | 15.20                   |
> | 77.69             | 64.13 ± 0.52                 | 17.45                   |
> | 81.24             | 62.20 ± 0.51                 | 23.44                   |
>
>
>
>
> These findings highlight that while the Gaussian diffusion model may approximate average porosity reasonably well near the mean, it fails to preserve structure and accuracy in the mean intensity away from this. More importantly, the lack of spatial coherence and the introduction of noise compromises the generated samples. This is especially important because transport properties in porous media are highly sensitive to such artifacts, small variations or noise can clog pores (percolation threshold) and significantly alter effective properties.
>
> **[Q1]** We thank the reviewer for the suggestion. However, respectfully, we are using standard notation. Since  $r$ is the transition rate (not a probability, but probability per unit time), we could not equate the $\mathbb{P}(\nu)$ to $r$ in the reviewer’s suggested notation (more about that on the answer to Reviewer tBZm's questions). Secondly, in the context of chemical reactions, $\nu$ is often to be interpreted as stoichiometric coefficients (which are not random variables) which prescribe the change of chemical species’ populations (in our context, we can think of discrete location $(x,y,c)$ as populations of three chemical species.) We would argue that the notation in the submitted version *is* the standard notation in existing technical literature, for example, see *“Solving the chemical master equation for monomolecular reaction systems and beyond: a Doi-Peliti path integral view”* by Vastola and *“Efficient analysis of stochastic gene dynamics in the non-adiabatic regime using piecewise deterministic Markov processes”* by Lin and Buchler. As such, we would like to retain our current model specification.  To improve the clarity of the manuscript we will explain the link between the transition rates and jump probabilities in the infinitesimal time limit.
>
> **[Q2]** We thank the reviewer for their suggestion. We have modified the sentence “The hyperparameters used can be found in Appendix H.” to “The hyperparameters used can be found in Table 3 in Appendix H.”
>
> **[Q3]** We thank the reviewer for the suggestions. We have inserted the dependence of $\bar{\nu}, x, y, c$ and explicitly annotated the sum and mean over these variables.
>
> **[Q4]** Thank you. We have fixed the typo.

---

> > ### Comment · Reviewer_RTj5 · 2025-08-04
> >
> > I thank the authors for their response, and for carrying out the requested experiment and corrections. I was mistaken in thinking that the notations are not standard. I will therefore raise my rating to accept.

---

### Official Review · Reviewer_tBZm · 2025-06-16

**Clarity:** 3
**Significance:** 3
**Originality:** 4
**Rating:** 5
**Confidence:** 2

**Summary:**

The paper proposes a discrete spatial diffusion (DSD) model, in which multiple particles are distributed in space and their total number is preserved throughout both the forward and reverse processes. This conservation property allows the model to control the number of generated particles by specifying the initial count in the reverse process. In both diffusion directions, each particle stochastically transitions to a neighboring region. A deep neural network (DNN) is employed to learn the transition probabilities at each time step of the reverse process. The training objective is conditioned on the observed data $I_0$ and is formulated analogously to standard diffusion models. The authors validate the effectiveness of the proposed approach through experiments on common image datasets (MNIST, CIFAR10, CelebA) and domain-specific data such as rock microstructures and lithium-ion battery electrodes.

**Questions:**

### Questions

* Regarding Eq. (1a):
    * This equation appears to describe the transition of a single particle in the forward process, with $r$ referred to as the transition rate. However, in Table 8, values such as $r = 120$ are used. What does it mean for the transition rate to be $r = 120$?
    * If $r$ represents the number of particles transitioning from $I_{x,y,c}$, how is the transition handled when $I_{x,y,c} < r$?
    * Does the model assume that transitions are made with equal probability to the four neighboring directions included in $\nu$ (or $\bar{\nu}$ in Eq. (3))?


### Comments
* In Appendix H, the authors state that “we found that our model was very robust with respect to the hyperparameters used.” It would be helpful to include a comparison of the forward and reverse processes under different values of $r$ (e.g., on MNIST), to illustrate this robustness more concretely.

* Throughout the paper, the abbreviation for the Inverse Heat Dissipation Model should be used consistently. It is currently inconsistent between “IHDM” and “IHD,” and should be unified.

**Ethical Concerns:**

["NO or VERY MINOR ethics concerns only"]

**Final Justification:**

The authors' rebuttal satisfactorily addressed my concern regarding the definition and role of the transition rate $r$. I appreciate the clear explanation, and this issue is now resolved.

There are no remaining concerns from my side. I maintain my original score, which reflects my overall positive assessment of the paper.

**Limitations:**

See weaknesses above.

**Paper Formatting Concerns:**

No concerns.

**Quality:**

3

**Strengths And Weaknesses:**

### Strengths
* A novel approach to modeling spatial distributions of particles using diffusion models while preserving the total number of particles.
* Experimental validation on real-world applications, including rock microstructures and lithium-ion battery electrodes.

### Weaknesses
* Certain aspects of the forward and reverse processes remain unclear (see Questions below).
* As noted by the authors, computational cost becomes a critical issue in high-intensity scenarios.

---

> ### Author Rebuttal · Authors · 2025-07-30
>
> Thank you for carefully reading of our work. We appreciate your thoughtful questions about the transition dynamics and your attention to detail regarding notation and clarity. Please find our detailed responses to your comments and questions below.
>
>
> **[Questions]**
>
> - We follow the standard model specification for continuous-time stochastic processes, for example, in Van Kampen’s *Stochastic Processes in Physics and Chemistry*, Gardiner’s *Stochastic Methods*, and Durrett’s *Essentials of stochastic processes*.
>
> - By definition, a transition rate is the probability of a potential “reaction” to take place per unit time, in the infinitesimal-time limit. For our forward process, $r=1000$ would mean that within time $\Delta t \downarrow 0$, a specific particle will move to its right (as well as left, up, and down) with a probability $p=1000 \Delta t$. The infinitesimal condition is imposed to eliminate multiple jump events, such as two consecutive jumps, which have negligible probability ($\mathcal{O}(\Delta t^2)$) in this infinitesimal time limit. Writing these operations with finite-time probability equations is far more complicated as it requires accounting for multiple-jump events even when the process is generated by simpler single-jump rates.  To improve the clarity of the manuscript we will explain this more carefully.
>
> - It is important to point out that in Eq. (1), $r$ stands for the transition for each of the particles located at $(x,y,c)$. Thus, even if there is only one particle, the transition rate is still $r=1000$. Since the transition rate is probability of a reaction to take place per unit time, we cannot compare its numerical value directly to the particle number: Even if we have only one particle at that location, it is fine to have $r=1000$; it just means that the particle has a higher propensity to make a move. In addition, we remark that when there are $I_{x,y,c}$ particles, the *total transition rate* would be $r \times I_{x,y,c}$ due to independence between the particles; in this case, it states that within time $\Delta t \downarrow 0$, *one* of the $I_{x,y,c}$ particles will move to its right (as well as left, up, and down) with a probability $p=1000 I_{x,y,c} \Delta t$.
>
> - In terms of the choice of $r$, we chose it so that at time $t=1$, the spatial diffusion process ensures sufficient mixing to a random configuration (so easy to sample during inference; see for example the SSIM metric in the Appendix). We remark that $r$ and terminal time (1 in our case) are co-design choices. We heuristically designed the noise scheduler using the SSIM principle. We observed that the choice of the noise scheduler does not seem to significantly affect the performance of the model.
>
> - Yes, our model uses transitions that are made with equal probability to the four neighboring directions. This is because the particle’s transition rates to each of the directions are equal. Another way to characterize our forward process is that it is a spatially symmetric random walk. That said, we also carried-out experiments with no-flux boundary conditions (reported on Table 3) where the particles cannot transition periodically across the edge of the images. We did not identify discernible differences between the two approaches, showing the robustness of the spatial diffusion formulation.
>
> **[Comments]**
>
> - Thank you for pointing out the inconsistency in the use of the acronym. We have reviewed and corrected all occurrences to ensure consistent and accurate usage throughout the manuscript.

---

> > ### Comment · Reviewer_tBZm · 2025-08-04
> >
> > Thank you for the clarification regarding the transition rate $r$. I hope these points will be clearly explained in the final version of the paper. At this point, I would prefer to maintain my current score.

---

### Official Review · Reviewer_n1in · 2025-06-28

**Clarity:** 3
**Significance:** 3
**Originality:** 3
**Rating:** 5
**Confidence:** 4

**Summary:**

This paper proposes a novel variant of diffusion models that preserves particle counts throughout both the forward and reverse processes, which is a property desired for scientific applications where discrete quantity conservation is essential. The key idea is to reformulate the standard Gaussian corruption process as a continuous-time, discrete-state jump stochastic process that operates directly in the discrete spatial domain. The model’s effectiveness is demonstrated through experiments on two scientific applications, with additional validation conducted on standard image benchmarks for computer vision tasks.

**Questions:**

Please see the weaknesses mentioned above.

**Ethical Concerns:**

["NO or VERY MINOR ethics concerns only"]

**Final Justification:**

Keep my positive rating after rebuttal.

**Limitations:**

Yes.

**Paper Formatting Concerns:**

No.

**Quality:**

4

**Strengths And Weaknesses:**

**S1:** The problem addressed is niche but compelling in the context of scientific applications. The paper is well-structured and clearly presented, with both the motivation and core technical contributions effectively communicated.

**S2:** The Markovian jumping process is an interesting and intuitive idea to achieve the preservation of intensity from the formulation level, thus presenting a rather elegant way to solve the challenge.

**S3:** The experiments effectively demonstrate the benefits of the proposed intensity-preserving property, even though this characteristic may not appear immediately critical for general vision tasks.

**W1:** As acknowledged in the limitations section, the proposed DSD framework introduces significant computational overhead compared to standard Gaussian diffusion models, both during training and inference. Given that Gaussian diffusion is already computationally intensive, this further limits the practical adoption of DSD in mainstream vision applications.

**W2:** The experimental setup and evaluation protocols, particularly for the scientific use cases, remain somewhat unclear. For instance, is the goal to synthesize images of subsurface rock microstructures and lithium-ion electrodes from randomly initialized images, as outlined in Algorithm 2? If so, the inference process appears to require predefined condition numbers and intensity values to initiate generation. How feasible or practical is this requirement within real-world scientific workflows? Similarly, on the evaluation side, while the proposed DSD demonstrates improved intensity preservation, are the generated scientific samples rigorously valid within their respective domains? A more detailed discussion on the scientific fidelity and downstream utility of the generated data would strengthen the work.

Minors: There are some formatting inconsistencies across subtitles, for example, some section headers begin with bolded first letters while others do not.

---

> ### Author Rebuttal · Authors · 2025-07-30
>
> Thank you for the detailed and constructive feedback, we're glad the motivation and elegance of the approach came through, and appreciate the opportunity to clarify our scientific evaluation protocols.
>
>
> **[W2]** Thank you for pointing this out. We apologize that aspects of the scientific dataset evaluation were unclear. For the human-centric datasets, we initialized inference from a random distribution to demonstrate our model’s capacity to generate human-centric RGB images. However, for the scientific domains (here, rock and battery microstructures), the inference allows for more practical use cases. After training, our framework can interpolate across intensity values not seen during training, enabling the exploration of intermediate or extrapolated physical states. For instance, in Appendix F.3 (MNIST), we show that the model produces coherent, semantically valid outputs even under out-of-distribution intensity conditions, suggesting learned representations that generalize and reflect meaningful physical structure.
>
> The ultimate goal of scientific applications is to generate microstructures conditioned on specific parameters for downstream analysis. In earth sciences, the types of samples we present in this work are immensely valuable, not only because acquiring them can cost millions of dollars, but because they offer a window through which we can observe and measure the complexity of the geology below us. Extracting as much information as possible from these scarce datasets, and then being able to generate new realizations, opens up avenues like performing uncertainty quantification, or interpolate to physical regimes that were not directly observed during sampling.
>
> Practically, in the case of rock microstructures, the generated samples could inform decisions in applications such as aquifer remediation, CO₂ sequestration, or hydrogen storage. In battery science, the aim is to generate novel electrode morphologies that are optimized for energy storage performance, subject to physical and material constraints (largely driven by economics). With these in mind, in real-world workflows, summary descriptors like porosity (for rocks) can often be estimated without expensive 3D x-ray imaging for example, through bulk measurements, indirect methods at the field (e.g., well logs), rapid and inexpensive lab-based porosity tests, or conventional 2D imaging, making it feasible to condition generation on realistic parameters. This opens up avenues for generating physically plausible realizations consistent with known constraints.
>
> We also conducted rigorous domain-specific evaluations. For the rock samples, we computed pore size distributions and two-point correlation functions using PoreSpy[1]. These showed excellent agreement between real and generated samples (Appendix F.2, Fig. 16). For battery electrodes, we evaluated interface length, triple-phase boundary, and relative diffusivity using TauFactor[2] (Appendix G.2, Fig. 19), all of which quantitatively confirm the structural and functional fidelity of our generated images. Additionally, we report a significantly improved FID score (0.9 vs. 18.1 in [3]), although we acknowledge FID’s limitations for binary images. These analyses support the scientific validity and potential utility of DSD samples in downstream modeling/design/ and uncertainty quantification tasks.
>
>  [1] Gostick, J. T., Khan, Z. A., Tranter, T. G., Kok, M. D., Agnaou, M., Sadeghi, M., & Jervis, R. (2019). PoreSpy: A python toolkit for quantitative analysis of porous media images. Journal of Open Source Software, 4(37), 1296.
>
> [2] Cooper, S. J., Bertei, A., Shearing, P. R., Kilner, J. A., & Brandon, N. P. (2016). TauFactor: An open-source application for calculating tortuosity factors from tomographic data. SoftwareX, 5, 203-210.
>
> [3] Kang-Hyun Lee and Gun Jin Yun. Microstructure reconstruction using diffusion-based generative models. Mechanics of Advanced Materials and Structures, 31(18):4443–4461, 2024.
>
> **[Minors]** Thank you, we have identified some inconsistencies in the capitalization used within section titles and have resolved the matter; hopefully this is the inconsistency to which you refer. We will make every effort to hunt down any other formatting problems, and if you have further issues to note then just let us know.

---

> > ### Comment · Reviewer_n1in · 2025-08-03
> >
> > I thank the authors for their rebuttal. It answered my questions, and I will keep my rating.

---

### Official Review · Reviewer_ADQ6 · 2025-06-29

**Clarity:** 3
**Significance:** 3
**Originality:** 4
**Rating:** 5
**Confidence:** 3

**Summary:**

The authors proposes Discrete spatial diffusion (DSD), a continuous time, discrete-state diffusion framework that preserves total image intensity (i.e., particle counts) in both forward and reverse processes by modeling intensity units as spatially correlated random walks. Their proposed method achieves competitive performance compared to continuous diffusion models (in the natural image domain), and demonstrates strong domain specific performance by accurately reconstructing porous rock microstructures and Li-ion battery electrode morphologies under strict intensity conservation.

**Questions:**

The custom jump-process and adaptive sampler adds engineering complexity into well established diffusion frameworks. Are there plans to release modular code or wrappers that would make DSD easy to plug into existing open source codebases?

**Ethical Concerns:**

["NO or VERY MINOR ethics concerns only"]

**Final Justification:**

My final justification is to keep my original score. On the matter of oral/spotlight, I believe a strong rating from all reviewers make it eligible for spotlight, but it might be focused on a very narrow field for an oral presentationl.

**Limitations:**

yes

**Paper Formatting Concerns:**

.

**Quality:**

4

**Strengths And Weaknesses:**

Strengths
1. The work has a clear problem motivation, addressing the need for generative models that strictly conserve total image intensity, which is needed for scientific and engineering applications where mass/energy preservation is non‐negotiable.
2. The proposed method guarantees strict mass conservation per color channel, enabling scientifically meaningful generation under hard constraints. They also rigorously justify their work through theory. The derivations of forward and reverse markov jump processes, noise scheduling via SSIM, and principled loss functions (rate matching) are clear.

Weaknesses
Their proposed method's forward and reverse sampling scales linearly with total intensity, leading to slower inference compared to vanilla diffusion models and potentially prohibits runtimes for high bit depth or very high resolution images. However, I acknowledge that these weaknesses are not within the scope of this work, and the authors also make an effor to acknowledge these weaknesses through engineering implementations and a section in their appendix (F.3) that directly addresses that their method remains computationally feasible.

Overall, this work is technically solid with high impact in generative modeling under conservation laws, well-substantiated theory, and strong domain specific results. Minor efficiency and usability tradeoffs do not outweigh its contributions.

---

> ### Author Rebuttal · Authors · 2025-07-30
>
> Thank you for the thoughtful and encouraging review, we're happy that the motivation and theoretical clarity came through.
>
> **[Q1]** We appreciate the comment. The codebase which we will release is relatively modular and designed using scipy, numpy, pytorch, and the scoremodels python package so as to best make use of existing tools and isolate the complexities associated with DSD in particular. We have routines for a) generating the Greens functions associated with the noising process, b) applying the noising process using dataloader workers, and c) sampling from a trained DSD network, which are where the additional engineering complexities lie within DSD. A user can easily swap in standard diffusion components, such as different architectures, and time (noise) schedules. Datasets in binarized or RGB formats can be incorporated with minimal effort, and it is possible to modify the integration parameters/strategies used during sampling. We will release the self-contained codebase on GitHub upon acceptance.

---

> > ### Comment · Reviewer_ADQ6 · 2025-08-04
> >
> > I thank the authors for addressing my question. Looking forward to the open source release. I would like to keep my score.

---

### Decision · Program_Chairs · 2025-09-17

**Decision:**

Accept (spotlight)

**Comment:**

The paper proposed discrete spatial diffusion (DSD), based on discrete-state jump process for image generation with total intensity preserved.  The proposed method performs comparably to the standard Gaussian diffusion on general image generation (although slower), and shows clear advantages in scientific applications including porous rock microstructures and lithium-ion battery electrodes.

All reviewers acknowledged impactful contributions, and some concerns, including clarity and lack of investigation, have been fully addressed in the rebuttal.   Reviewers recommended the paper to be a spotlight, because of high novelty on an important research area.

AC would suggest the following revision.  The original version first shows comparable result to the standard diffusion model in general image generation, and then show results on the scientific applications ONLY by the proposed method.  Consequently, readers cannot find the real advantage of the proposed method.   My suggestion is to put more weight on the scientific applications, and compare the proposed method with the standard diffusion models.  Then, reader will clearly see what are the problems when the standard diffusion is used for those scientific applications, and that the proposed methods solve those problems.  I believe that this would make the story of the paper clearer, and more convincing.